 SHORT REPORT

# Protection of nascent DNA at stalled replication forks is mediated by phosphorylation of RIF1 intrinsically disordered region

Sandhya Balasubramanian[1,2†], Matteo Andreani[1,2†‡], Júlia Goncalves Andrade[1], Tannishtha Saha[1,2], Devakumar Sundaravinayagam[1], Javier Garzón[3§], Wenzhu Zhang[4], Oliver Popp[5], Shin-ichiro Hiraga[3], Ali Rahjouei[1], Daniel B Rosen[6#], Philipp Mertins[5], Brian T Chait[4], Anne D Donaldson[3], Michela Di Virgilio[1,7*]

[1]Laboratory of Genome Diversification & Integrity, Max Delbrück Center for Molecular Medicine in the Helmholtz Association, Berlin, Germany; [2]Freie Universität Berlin, Berlin, Germany; [3]Institute of Medical Sciences, University of Aberdeen, Foresterhill, Aberdeen, United Kingdom; [4]Laboratory of Mass Spectrometry and Gaseous Ion Chemistry, The Rockefeller University, New York, United States; [5]Proteomics Platform, Max Delbrück Center for Molecular Medicine in the Helmholtz Association and Berlin Institute of Health, Berlin, Germany; [6]Laboratory of Molecular Immunology, The Rockefeller University, New York, United States; [7]Charité-Universitätsmedizin Berlin, Berlin, Germany

**\*For correspondence:**
michela.divirgilio@mdc-berlin.de

†These authors contributed equally to this work

**Present address:** ‡Tacalyx GmbH, Bayer CoLaborator, Berlin, Germany; §Adrestia Therapeutics Ltd., Babraham Research Campus, Cambridge, United Kingdom; #Department of Radiation Oncology, Dana-Farber Cancer Institute, Brigham and Women's Hospital, Boston, United States

**Abstract** RIF1 is a multifunctional protein that plays key roles in the regulation of DNA processing. During repair of DNA double-strand breaks (DSBs), RIF1 functions in the 53BP1-Shieldin pathway that inhibits resection of DNA ends to modulate the cellular decision on which repair pathway to engage. Under conditions of replication stress, RIF1 protects nascent DNA at stalled replication forks from degradation by the DNA2 nuclease. How these RIF1 activities are regulated at the post-translational level has not yet been elucidated. Here, we identified a cluster of conserved ATM/ATR consensus SQ motifs within the intrinsically disordered region (IDR) of mouse RIF1 that are phosphorylated in proliferating B lymphocytes. We found that phosphorylation of the conserved IDR SQ cluster is dispensable for the inhibition of DSB resection by RIF1, but is essential to counteract DNA2-dependent degradation of nascent DNA at stalled replication forks. Therefore, our study identifies a key molecular feature that enables the genome-protective function of RIF1 during DNA replication stress.

## Editor's evaluation

This paper reports a novel regulatory mechanism that modulates RIF1 function during the DNA replication stress response. The authors identify three residues within the mouse RIF protein that can be phosphorylated in an ATM/ATR-dependent manner. Interestingly, this phosphorylation is dispensable for the ability of RIF1 to limit double-strand break resection but is required to protect stalled replication forks. The results are of clear interest for the field of DNA replication and repair.

## Introduction

Control of DNA processing is a crucial determinant for the preservation of genome stability during both DNA repair and DNA replication. In the context of DNA double-strand break (DSB) repair, nucleolytic processing of DNA ends acts as a key defining step in the regulation of repair pathway choice (*Chapman et al., 2012*; *Scully et al., 2019*). Extensive 5′ to 3′ resection of DSBs inhibits repair by nonhomologous end joining (NHEJ) but is a prerequisite for homology-dependent repair processes (homologous recombination [HR] and alternative end joining [A-EJ]) (*Chang et al., 2017*; *Symington, 2016*). These pathways are differentially engaged to mediate physiological DSB repair according to the cellular context, cell cycle phase, and type of break (*Chang et al., 2017*; *Chapman et al., 2012*; *Scully et al., 2019*). As a result, dysregulated DSB end processing can lead to unproductive or aberrant repair reactions with dramatic consequences at both cellular and systemic levels, as evidenced during repair of programmed DSBs in B lymphocytes undergoing class switch recombination (CSR) and of stochastic DNA replication-associated breaks in BRCA1-mutated cells.

Immunoglobulin (Ig) CSR is the process occurring in mature B lymphocytes that enables the formation of different Ig classes or isotypes, thus diversifying the effector component of immune responses (*Methot and Di Noia, 2017*). At the molecular level, CSR is mediated by a deletional recombination reaction at the Ig heavy chain locus (*Igh*) that replaces the constant (C) gene for the basal IgM isotype with one of the downstream C genes encoding a different Ig class (*Yewdell and Chaudhuri, 2017*). The reaction is initiated by the formation of multiple programmed DSBs at internally repetitive DNA stretches, known as switch (S) regions, preceding the recombining C regions (*Saha et al., 2021*). Productive CSR events occur via protection of DSBs from nucleolytic resection, which enables NHEJ-mediated inter-S-region repair (*Boboila et al., 2012*; *Saha et al., 2021*). Defects in DSB end protection lead to unscheduled processing of S region breaks, which, combined with the close break proximity and the repetitive nature of these DNA stretches, favors local, hence unproductive, intra-S-region recombination reactions, and results in immunodeficiency (*Boersma et al., 2015*; *Chapman et al., 2013*; *Dev et al., 2018*; *Di Virgilio et al., 2013*; *Escribano-Díaz et al., 2013*; *Findlay et al., 2018*; *Ghezraoui et al., 2018*; *Gupta et al., 2018*; *Hakim et al., 2012*; *Ling et al., 2020*; *Noordermeer et al., 2018*; *Panchakshari et al., 2018*; *Reina-San-Martin et al., 2007*; *Xu et al., 2015*; *Yamane et al., 2013*).

Conversely, extensive processing is essential for repair of DNA replication-associated breaks, which employs HR as the physiological repair pathway (*Scully et al., 2019*; *Symington, 2016*). In this context, the HR protein BRCA1 specifically counteracts DSB end protection, thus enabling resection and HR (*Bunting et al., 2010*; *Tarsounas and Sung, 2020*). Absence of BRCA1 causes persistent protection of DNA replication-associated DSBs, which interferes with their physiological repair by HR (*Bunting et al., 2010*). As a result, cells accumulate unrepaired DSBs and aberrant NHEJ-mediated chromosome fusions known as radials (*Bouwman et al., 2010*; *Bunting et al., 2010*). The increased levels of genome instability are responsible for the lethality of BRCA1-mutated cells and mouse models (*Tarsounas and Sung, 2020*). Defects in DSB end protection can relieve the inhibitory brake on resection in BRCA1-mutated cells and partially rescue HR, genome stability, and viability (*Bouwman et al., 2010*; *Bunting et al., 2010*; *Cao et al., 2009*; *Chapman et al., 2013*; *Dev et al., 2018*; *Escribano-Díaz et al., 2013*; *Feng et al., 2013*; *Findlay et al., 2018*; *Ghezraoui et al., 2018*; *Gupta et al., 2018*; *Noordermeer et al., 2018*; *Xu et al., 2015*; *Zimmermann et al., 2013*).

Recently, pathways that counteract the nucleolytic degradation of nascent DNA at replication forks have proven to be crucial to maintain genome stability under conditions of replication stress (*Pasero and Vindigni, 2017*; *Rickman and Smogorzewska, 2019*; *Schlacher et al., 2011*). Replication fork reversal is the mechanism that converts a classic three-way junction fork into a four-way junction structure via the annealing of the newly synthesized complementary DNA strands and the re-annealing of the parental strands (*Neelsen and Lopes, 2015*). This process, which results in the formation of a fourth regressed arm, appears to have a stabilizing effect on forks stalled as a consequence of DNA replication stress (*Liao et al., 2018*). However, reversed forks can act as the entry point for various DNA nucleases, and unrestrained processing of the newly replicated DNA in the absence of protective factors leads to accumulation of DNA breaks and hypersensitivity to replication stress-inducing agents (*Cortez, 2015*; *Neelsen and Lopes, 2015*).

The multifunctional protein RIF1 has emerged as a key regulator of DNA processing. During repair of DSBs, RIF1 acts in the 53BP1/Shieldin-mediated cascade that inhibits resection of DNA ends

(*Chapman et al., 2013*; *Di Virgilio et al., 2013*; *Escribano-Díaz et al., 2013*; *Feng et al., 2013*; *Zimmermann et al., 2013*). As a consequence, ablation of RIF1 in mature B cells severely impairs NHEJ repair of CSR DSBs and leads to immunodeficiency in mouse models (*Chapman et al., 2013*; *Di Virgilio et al., 2013*; *Escribano-Díaz et al., 2013*). Conversely, deletion of RIF1 in BRCA1-deficient cells partially restores resection and HR-dependent repair of DNA replication-associated breaks, and reduces genome instability and cell lethality of this genetic background (*Chapman et al., 2013*; *Escribano-Díaz et al., 2013*; *Feng et al., 2013*; *Zimmermann et al., 2013*). Furthermore, recent studies have uncovered a DNA protective role of RIF1 during replication stress (*Ray Chaudhuri et al., 2016*; *Garzón et al., 2019*; *Mukherjee et al., 2019*). RIF1 is recruited to stalled DNA replication forks and protects newly synthesized DNA from processing by the DNA2 nuclease (*Garzón et al., 2019*; *Mukherjee et al., 2019*). Loss of RIF1 leads to increased degradation of nascent DNA at reversed forks and exposure of under-replicated DNA and genome instability (*Ray Chaudhuri et al., 2016*; *Garzón et al., 2019*; *Mukherjee et al., 2019*).

Despite the multiple contributions of RIF1 in the regulation of DNA processing and the consequences on the preservation of genome integrity, very little is known about the post-translational control of RIF1 activities in these contexts. Furthermore, although the DSB resection inhibitory function of RIF1 has been the objective of extensive investigation, little information is available about how its DNA replication fork protective role is regulated. In this study, we report the identification of a cluster of conserved SQ motifs within mammalian RIF1 that is phosphorylated in actively proliferating B lymphocytes. Abrogation of these phosphorylation events does not affect RIF1's ability to inhibit DSB resection but severely impairs RIF1-mediated protection of stalled DNA replication forks.

## Results

### A conserved cluster of SQ sites in RIF1 intrinsically disordered region is phosphorylated in activated B cells

RIF1 is a large protein of almost 2500 amino acids in mammalian cells (*Figure 1—source data 1*) with no known enzymatic activity. While information about RIF1 structural organization is limited, analyses of RIF1 homologs across species identified two motifs that are highly conserved from yeast to mammals: the N-terminal *H*untingtin, *E*longation factor 3, *A* subunit of protein phosphatase 2A, and *T*or1 (HEAT) repeats, and the SILK-RVxF motif, whose sequence location shifted from the N-terminus to the C-terminal end during the evolution of unicellular to multicellular organisms (*Figure 1A*; *Sreesankar et al., 2012*; *Xu et al., 2010*). In vertebrates, RIF1 also exhibits a conserved C-terminal domain with a tripartite structure (*Figure 1A*; *Xu et al., 2010*). The region spanning between these N- and C-terminal motifs is poorly conserved and is characterized by a high degree of intrinsic disorder (*Figure 1A*). Additionally, RIF1 contains multiple serine-glutamine/threonine-glutamine (SQ/TQ) motifs, which are consensus sites for phosphorylation by the DNA damage response kinases ATM and ATR (*Blackford and Jackson, 2017*; *Figure 1A*, *Figure 1—source data 2*).

To identify post-translational modifications (PTMs) that modulate RIF1 functions in the maintenance of genome stability, we took advantage of the *I*sotopic *D*ifferentiation of *I*nteractions as *R*andom or *T*argeted (I-DIRT) experiment that we recently performed to define RIF1 interactome in mature B lymphocytes activated to differentiate ex vivo (*Delgado-Benito et al., 2018*). In addition to experiencing programmed DSB formation and repair during Ig CSR, activated B cells undergo a proliferative burst that renders them susceptible to DNA replication stress and damage (*Figure 1B*). Furthermore, activated B cells express considerably higher levels of RIF1 than their mouse embryonic fibroblast (MEF) counterparts (*Figure 1C*). The I-DIRT approach employed primary cultures of splenocytes from mice harboring a FLAG-2xHA-tagged version of RIF1 (RIF1[FH], *Cornacchia et al., 2012*; *Delgado-Benito et al., 2018*), which is expressed at physiological levels (*Figure 1C*; *Delgado-Benito et al., 2018*). For the RIF1 I-DIRT experiment, activated splenocytes cultures were also irradiated, which would simultaneously increase the level and broaden the range of DNA damage-induced PTMs (*Figure 1D*; *Delgado-Benito et al., 2018*). Furthermore, αFLAG-mediated pull-down of RIF1 was performed under conditions that preserved bona fide protein interactions and native complex formation (*Figure 1D*; *Delgado-Benito et al., 2018*). The RIF1 I-DIRT experiment generated a list of high-confidence interactor candidates with functions ranging from DSB repair to transcriptional regulation of gene expression (*Figure 1D*; *Delgado-Benito et al., 2018*). Moreover, a differential filtering

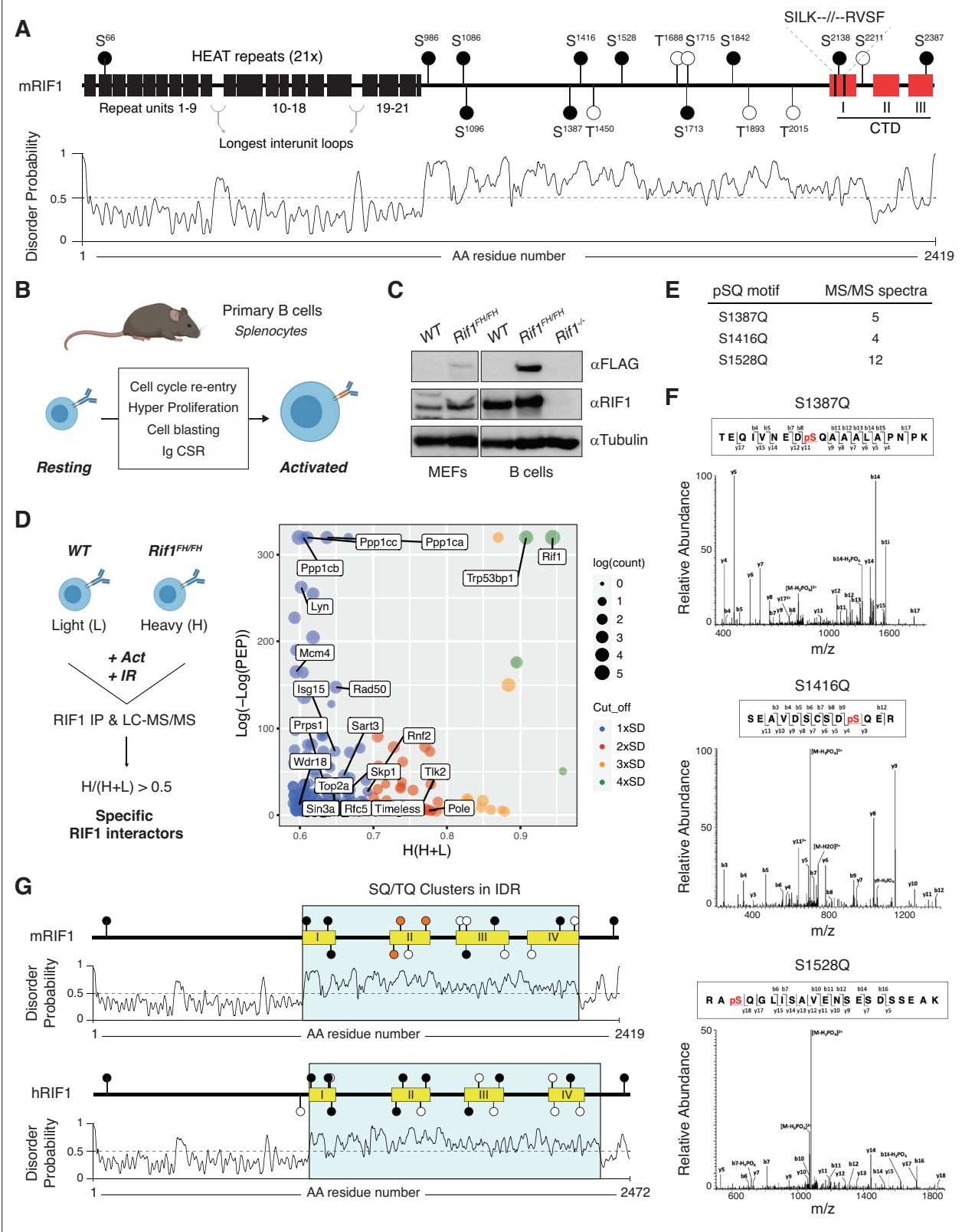

**Figure 1.** A conserved cluster of serine-glutamine (SQ) motifs within RIF1 intrinsically disordered region (IDR) is phosphorylated in activated B lymphocytes. (**A**) Top: schematic representation of mammalian RIF1 domains and motifs. The scheme refers to the canonical sequence for mouse RIF1 (mRIF1, isoform 1, 2419 amino acids, UniProt entry Q6PR54-1). Filled and empty circle symbols represent conserved and nonconserved SQ/threonine-glutamine (TQ) motifs, respectively, between mRIF1 and human RIF1 (hRIF1, isoform 1, 2472 amino acids, UniProt entry Q5UIP0-1) (see also **Figure**

*Figure 1 continued on next page*

*Figure 1 continued*

*1—source data 1* and *Figure 1—source data 2*). CTD: carboxyl-terminal domain. Bottom: disorder profile plot of mRIF1 as determined by *Protein DisOrder prediction System* (PrDOS). (**B**) Schematic representation of key cellular changes and processes induced by the activation of mature B lymphocytes. Ig CSR: immunoglobulin class switch recombination. (**C**) Western blot analysis of whole-cell extracts from mouse embryonic fibroblasts (MEFs) and primary B cells derived from mice of the indicated genotypes. For each depicted antibody staining, the left and right blots represent noncontiguous portions of the same gel and film exposure. *Rif1$^{-/-}$: Rif1$^{F/F}$Cd19$^{Cre/+}$*. (**D**) Left: schematic representation of RIF1 *Isotopic Differentiation of Interactions as Random or Targeted* (I-DIRT) in primary cultures of B cells. Light (L): light media; heavy (H): heavy media; Act: activation; IR: ionizing radiation; LC-MS/MS: liquid chromatography-tandem mass spectrometry. Right: graph depicting the distribution of identified RIF1 I-DIRT proteins as a function of their H/(H+L) ratio and posterior error probability (PEP) (data from *Delgado-Benito et al., 2018*). Only proteins with PEP ≤ 10$^{-4}$ were included in the graph. SD: standard deviation units (0.10) from the mean of the distribution (0.49); Count: number of peptides identified per protein. (**E**) Number of MS/MS spectra identified for the indicated phosphorylated SQ (pSQ) motif-containing peptides in different RIF1 I-DIRT preparations. (**F**) Representative MS/MS spectra of the RIF1 peptides encompassing phosphorylated residues S$^{1387}$, S$^{1416}$, and S$^{1528}$. (**G**) Schematic representation of SQ/TQ motif clusters in the IDRs of mouse and human RIF1, which were defined by the PrDOS disorder profile plots (*Ishida and Kinoshita, 2007*). Orange filled symbols represent the conserved S$^{1387}$, S$^{1416}$, and S$^{1528}$ residues identified as phosphorylated SQ motifs in mRIF1.

The online version of this article includes the following source data for figure 1:

**Source data 1.** List of RIF1 protein homologs across representative species from the Animalia and Fungi kingdoms.

**Source data 2.** Alignment of peptides containing SQ/TQ motifs conserved between mouse and human RIF1 proteins across representative species from the Animalia and Fungi kingdoms.

**Source data 3.** Original file for the Western blot analysis in *Figure 1C* (anti-FLAG, anti-RIF1, and anti-tubulin).

**Source data 4.** PDF containing *Figure 1C* and original scans of the relevant Western blot analysis (anti-FLAG, anti-RIF1, and anti-tubulin) with highlighted bands and sample labels.

**Source data 5.** Excel file containing output results of MaxQuant analysis for the potential RIF1 interactors for the graph in *Figure 1D*.

criteria analysis uncovered an extended network of factors contributing to DNA replication initiation, elongation, and fork protection (*Figure 1D*). Altogether, these observations indicate that activated B cells provide an ideal model system to probe for RIF1 multiple biological functions and prompted us to re-evaluate RIF1 I-DIRT datasets for potentially relevant PTMs.

Analysis of post-translationally modified RIF1 peptides from different I-DIRT preparations revealed phosphorylation to be the predominant PTM, with the majority of phosphoresidues being serines followed by either a proline or a glutamic acid (SP or SE) (data not shown). Among all SQ/TQ motifs present in mouse RIF1, S$^{1387}$Q, S$^{1416}$Q, and S$^{1528}$Q were reproducibly found to be phosphorylated across independent RIF1 I-DIRT datasets (*Figure 1E and F*). These SQ motifs exhibit a relatively high degree of conservation across species (*Figure 1—source data 1* and *Figure 1—source data 2*). More interestingly, S$^{1387}$Q, S$^{1416}$Q, and S$^{1528}$Q (S$^{1403}$Q, S$^{1431}$Q, and S$^{1542}$Q in hRIF1) are located in close proximity to each other and form a defined cluster of SQ sites in the IDR of both mouse and human RIF1 (IDR-CII SQs) (*Figure 1G*).

We concluded that in activated B lymphocytes, RIF1 is phosphorylated at a conserved cluster of SQ motifs within its IDR.

## A genetic engineering-amenable B cell model system for the assessment of DSB end protection outcomes

Phosphorylation of residues within IDRs has been reported to affect protein functions in a variety of cellular contexts (*Bah and Forman-Kay, 2016*; *Wright and Dyson, 2015*). Given the conservation, proximity, and IDR location of S$^{1387}$Q, S$^{1416}$Q, and S$^{1528}$Q motifs, as well as their identification as phosphoresidues in I-DIRT pull-downs, we decided to assess the contribution of the IDR-CII SQ phosphorylation to the regulation of RIF1 activities in DNA repair. RIF1 inhibits resection of DSBs downstream 53BP1 during both aberrant repair of DNA replication-associated DSBs in the absence of BRCA1 and physiological end joining of CSR breaks in G1 in B cells (*Chapman et al., 2013*; *Di Virgilio et al., 2013*; *Escribano-Díaz et al., 2013*; *Feng et al., 2013*; *Zimmermann et al., 2013*). Therefore, to determine if phosphorylation of the IDR-CII modulates RIF1's role in DSB end protection, we monitored both types of repair in cells expressing phosphomutant RIF1.

To assess for aberrant (radial chromosome formation) and physiological (CSR) repair events in the same cellular context, we opted to perform our analysis in HR-deficient, yet CSR-proficient, CH12 cells bearing hypomorphic *Brca1* mutations (*Figure 2A*). CH12 is a well-characterized mouse B cell lymphoma line that recapitulates the molecular mechanism and regulation of CSR (*Nakamura et al.,*

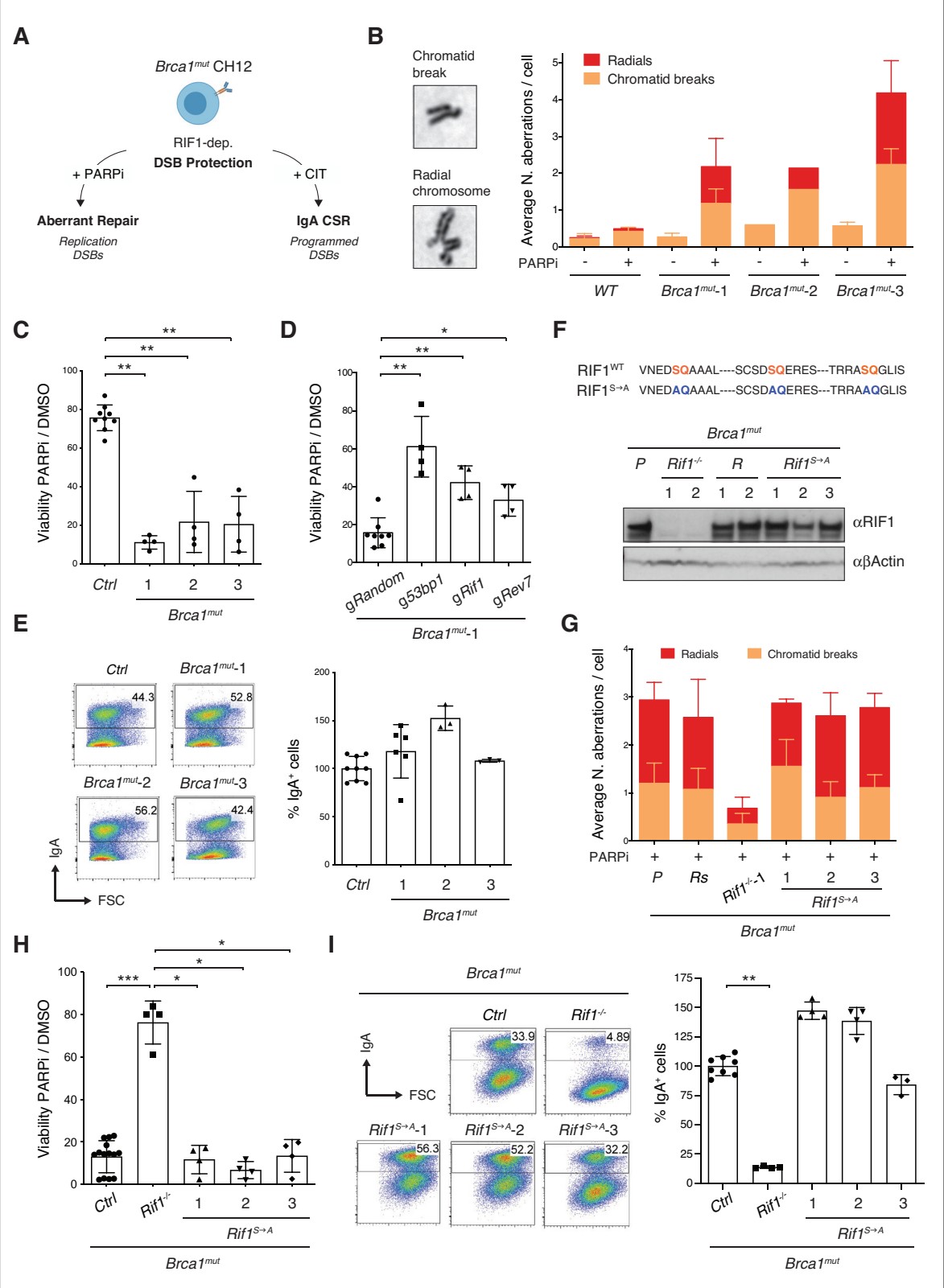

**Figure 2.** Phosphorylation of RIF1 at the conserved IDR-CII serine-glutamine (SQ) motifs is dispensable for its roles in double-strand break (DSB) end protection. (**A**) Schematic representation of BRCA1-deficient CH12 model system's versatility to investigate both pathological and physiological consequences of RIF1-mediated DSB end protection. CIT: αCD40, *IL-4*, and *TGF*β B cell activation cocktail. (**B**) Left: representative images of chromosomal aberrations typically associated with homologous recombination (HR) deficiency (chromatid breaks and radial chromosomes). Right:

*Figure 2 continued on next page*

*Figure 2 continued*

graph summarizing the average number of chromosomal aberrations in the parental CH12 cell line (WT sample) and selected *Brca1^mut* clonal derivatives following 1 µM PARPi treatment for 24 hr from two independent experiments (n = 50 metaphases analyzed per genotype). Breakdown of the same data into actual number of aberrations per cell is shown for one experimental repeat in *Figure 2—figure supplement 1C*. (**C**) Residual viability of *Brca1^mut* CH12 cell lines after treatment with 1 µM of PARPi versus DMSO (mock treatment control) for 72 hr. Residual viability was calculated as percentage of cell viability of PARPi- over DMSO-treated cultures. Graph summarizes four independent experiments per *Brca1^mut* clonal derivative. The control (*Ctrl*) samples comprise parental WT CH12 cells and clonal cell lines generated by targeting CH12 cells with gRNAs against random sequences not present in the mouse genome (validated Random clones, *Brca1^mutR*). (**D**) Residual viability of *Brca1^mut*-1 CH12 cells nucleofected with random gRNAs (*Random*), or *53bp1*, *Rif1*, and *Rev7*, and treated for 72 hr with 1 µM of PARPi versus DMSO. Graph summarizes four independent experiments. (**E**) Left: representative flow cytometry plots measuring class switch recombination (CSR) to IgA in activated cell lines of the indicated genotype. Right: summary graph for at least three independent experiments per *Brca1^mut* cell line, with CSR% levels within each experiment normalized to the average of controls (parental WT CH12 and one Random clone), which was set to 100. (**F**) Top: amino acid sequence in the IDR-CII SQ region of WT and S→A-mutated RIF1 protein. Bottom: Western blot analysis of whole-cell extracts from independent cells lines of the indicated genotypes (*Rif1^-/-*, control Random *R*, and *Rif1^{S→A}*, all generated on the parental [P] *Brca1^mut*-1 cell line background, henceforth indicated as *Brca1^mut*). (**G**) Graph summarizing the average number of chromosomal aberrations in cells of the indicated genotypes following 1 µM PARPi treatment for 24 hr with each *Brca1^mutRif1^{S→A}* cell line tested twice over three independent experiments (n = 50 metaphases analyzed per genotype). Control samples include the parental *Brca1^mut*-1 cell line (P) and a derivative *Brca1^mutR* clone. (**H**) Residual viability of *Brca1^mutRif1^{S→A}* cell lines after treatment with 1 µM of PARPi versus DMSO for 72 hr. Graph summarizes four independent experiments per *Brca1^mutRif1^{S→A}* clonal derivative. The control (*Ctrl*) samples comprise parental *Brca1^mut*-1 cells and *Brca1^mutR* clones. (**I**) Left: representative flow cytometry plots measuring CSR to IgA in activated cell lines of the indicated genotype. Right: summary graph for four independent experiments, with CSR% levels within each experiment normalized to the average of controls (parental *Brca1^mut*-1 and one Random clone), which was set to 100. Significance in panels (**C**), (**D**), (**H**), and (**I**) was calculated with the Mann–Whitney *U*-test, and error bars represent SD. *p≤0.05; **p≤0.01; ***p≤0.001.

The online version of this article includes the following source data and figure supplement(s) for figure 2:

**Source data 1.** Original file for the Western blot analysis in *Figure 2F* (anti-RIF1).

**Source data 2.** Original file for the Western blot analysis in *Figure 2F* (anti-β-actin).

**Source data 3.** PDF containing *Figure 2F* and original scans of the relevant Western blot analysis (anti-RIF1 and anti-β-actin) with highlighted bands and sample labels.

**Figure supplement 1.** BRCA1-mutated CH12 cell lines recapitulate the genomic instability of BRCA1 deficiency.

**Figure supplement 1—source data 1.** Original image of WT metaphase spreads in *Figure 2—figure supplement 1C*.

**Figure supplement 1—source data 2.** Original image of *Brca1^mut*-1 metaphase spreads in *Figure 2—figure supplement 1C*.

**Figure supplement 1—source data 3.** Original image of *Brca1^mut*-2 metaphase spreads in *Figure 2—figure supplement 1C*.

**Figure supplement 1—source data 4.** Original image of *Brca1^mut*-3 metaphase spreads in *Figure 2—figure supplement 1C*.

**Figure supplement 2.** Generation of RIF1-mutant CH12 cell lines on a BRCA1-mutant background.

**Figure supplement 2—source data 1.** Original file for the Western blot analysis in *Figure 2—figure supplement 2C* (anti-RIF1 and anti-β-actin).

**Figure supplement 2—source data 2.** PDF containing *Figure 2—figure supplement 2C* and original scans of the relevant Western blot analysis (anti-RIF1 and anti-β-actin) with highlighted bands and sample labels.

**Figure supplement 2—source data 3.** Original file for the diagnostic digestion in *Figure 2—figure supplement 2G*.

**Figure supplement 2—source data 4.** PDF containing *Figure 2—figure supplement 2G* and original files of the relevant diagnostic digestion with highlighted bands and sample labels.

**Figure supplement 2—source data 5.** Original file for the Western blot analysis in *Figure 2—figure supplement 2H* (anti-pRPA long).

**Figure supplement 2—source data 6.** Original file for the Western blot analysis in *Figure 2—figure supplement 2H* (anti-pRPA short).

**Figure supplement 2—source data 7.** Original file for the Western blot analysis in *Figure 2—figure supplement 2H* (anti-RPA).

**Figure supplement 2—source data 8.** PDF containing *Figure 2—figure supplement 2H* and original scans of the relevant Western blot analysis (anti-pRPA and anti-RPA) with highlighted bands and sample labels.

*1996*). Furthermore, CH12 cells display a stable near-diploid genome that can be easily and efficiently manipulated by somatic gene targeting (*Delgado-Benito et al., 2020*; *Delgado-Benito et al., 2018*; *Sundaravinayagam et al., 2019*).

These features render CH12 the preferred model system over B cells isolated from the available BRCA1-mutated mouse models, which (1) are refractory to classic transfection methods, (2) do not allow for transduction-based reconstitution studies of large proteins like RIF1, and (3) whose primary nature precludes genetic manipulation for knock-in generation.

To generate BRCA1-mutated CH12 cells able to support CSR, we introduced in-frame deletions specifically within exon 11 of the *Brca1* gene (*Björkman et al., 2015*; *Bunting et al., 2010*; *Callen*

*et al., 2013*). Targeted deletion of *Brca1* exon 11 in mice results in the expression of a splice variant (BRCA1- Δ11) that preserves intact N-terminal RING finger domain and C-terminal BRCT repeats but lacks key motifs that are essential for BRCA1 functions (*Evers and Jonkers, 2006*; *Xu et al., 2001*; *Xu et al., 1999*). BRCA1-Δ11-expressing B cells exhibit genome instability because of impaired HR but undergo CSR as proficiently as WT cells (*Bunting et al., 2010*; *Callen et al., 2013*). We employed two different nickase gRNA pairs directed towards the 5′- region of the exon (*Figure 2—figure supplement 1A*). All analyzed clones bore in-frame deletions, which are indicative of internally deleted, hypomorphic BRCA1 mutants (*Brca1ᵐᵘᵗ*, *Figure 2—figure supplement 1B*, and data not shown). To functionally confirm the partial loss of BRCA1 function, we assessed the levels of chromosomal aberrations in response to treatment with the PARP inhibitor olaparib (PARPi). PARPi increases the load of DNA replication-associated breaks, and in BRCA1-deficient backgrounds it triggers the accumulation of chromatid breaks and radial chromosomes (*Farmer et al., 2005*). These aberrations are caused by the inability to engage physiological repair by HR, in part because of suppressed DSB end resection (*Bouwman et al., 2010*; *Bunting et al., 2010*). The resulting genome instability is responsible for the increased cell lethality associated with PARPi treatment in this genetic background (*Farmer et al., 2005*; *Rottenberg et al., 2008*). Analysis of metaphase spreads revealed that in contrast to control cells PARPi-treated *Brca1ᵐᵘᵗ* CH12 cell lines accumulated chromatid breaks and radials with high frequency (*Figure 2B*, *Figure 2—figure supplement 1C*). Accordingly, all *Brca1ᵐᵘᵗ* cell lines displayed reduced viability in the presence of PARPi compared to their wild-type counterparts (*Figure 2C*). We concluded that *Brca1ᵐᵘᵗ* CH12 cells exhibit genome instability-driven cell death following PARPi treatment.

Deletion of DSB end protection factors in BRCA1-deficient cells releases the inhibition on DNA end resection and partially rescues HR, genome stability, and, as a consequence, viability (*Bouwman et al., 2010*; *Bunting et al., 2010*; *Cao et al., 2009*; *Chapman et al., 2013*; *Dev et al., 2018*; *Escribano-Díaz et al., 2013*; *Feng et al., 2013*; *Findlay et al., 2018*; *Ghezraoui et al., 2018*; *Gupta et al., 2018*; *Noordermeer et al., 2018*; *Xu et al., 2015*; *Zimmermann et al., 2013*). Therefore, we monitored the consequences of ablating key components of the DSB end protection cascade in *Brca1ᵐᵘᵗ* CH12 cell lines. In-bulk targeting of RIF1 as well as of the up- and downstream pathway components 53BP1 and REV7, respectively, led to a significant rescue of viability in *Brca1ᵐᵘᵗ* cells (*Figure 2D*). Furthermore, *Brca1ᵐᵘᵗRif1⁻/⁻* clonal derivatives exhibited reduced levels of chromosomal aberrations and a marked increase in viability after PARPi treatment compared to *Brca1ᵐᵘᵗ* cells (*Figure 2G*, *Figure 2—figure supplement 2A–D*). We concluded that *Brca1ᵐᵘᵗ* CH12 cell lines recapitulate the RIF1-dependent genome instability and cell lethality typical of BRCA1-deficient backgrounds.

CH12 cells can be induced to undergo CSR to IgA with high efficiency after activation with αCD40, IL-4, and TGFβ (CIT cocktail, *Nakamura et al., 1996*). NHEJ repair of CSR breaks in CH12 mimics the molecular requirements of the physiological process in primary B cells. Accordingly, *Brca1ᵐᵘᵗ* CH12 cell lines were able to undergo stimulation-dependent CSR to levels comparable to WT CH12 cells (*Figure 2E*), whereas deletion of RIF1 in these cells dramatically impaired CSR (*Figure 2—figure supplement 2E*).

Altogether, these findings show that *Brca1ᵐᵘᵗ* CH12 cell lines allow for the investigation of both outcomes of RIF1-mediated DSB end protection: aberrant repair of DNA replication-associated DSBs and physiological end joining of CSR breaks.

## Phosphorylation of the IDR-CII SQ cluster is dispensable for RIF1's ability to inhibit DSB end resection

To investigate whether the phosphorylation status of the conserved IDR-CII SQ cluster is required for RIF1's ability to inhibit DSB end resection, we abrogated phosphorylation of $S^{1387}Q$, $S^{1416}Q$, and $S^{1528}Q$ motifs by serine to alanine substitutions in *Brca1ᵐᵘᵗ* CH12 cells via CRISPR-Cas9-mediated knock-in mutagenesis at the *Rif1* locus (*Figure 2F*, *Figure 2—figure supplement 2F and G*). This knock-in approach allows the characterization of the PTM-dependent regulation of RIF1 biological functions under physiological levels of protein expression.

Despite the expected HR deficiency of *Brca1ᵐᵘᵗ* cells, we obtained several clonal derivatives that harbored the desired mutations (*Brca1ᵐᵘᵗRif1^{S→A}*) and expressed wild-type levels of RIF1 (*Figure 2F*). To control for any clonality-related issue, we employed three independent clonal derivatives for all subsequent analyses.

We first asked whether preventing phosphorylation of the IDR-CII SQ cluster affected RIF1's ability to inhibit resection during repair of DNA replication-associated DSBs. To do so, we assessed chromosomal aberrations and viability following PARPi treatment. All $Brca1^{mut}Rif1^{S \to A}$ cell lines accumulated chromatid breaks and radial chromosomes to the same levels as the control $Brca1^{mut}$ genotype (*Figure 2G*) and were as sensitive to the treatment (*Figure 2H*). In contrast, $Brca1^{mut}Rif1^{-/-}$ cells exhibited the expected reduction in chromosomal aberrations and rescue of viability (*Figure 2G and H*). These results show that abrogation of phosphorylation events at the conserved cluster does not affect RIF1's role in promoting genome instability in BRCA1-deficient cells.

We next assessed the contribution of IDR-CII SQ cluster phosphorylation on CSR, which is dependent on RIF1's ability to protect CSR breaks against resection (*Chapman et al., 2013*; *Di Virgilio et al., 2013*; *Escribano-Díaz et al., 2013*). To this end, we stimulated control and $Brca1^{mut}Rif1^{S \to A}$ cell lines with αCD40, TGFβ, and IL-4, and monitored CSR efficiencies. Whereas $Brca1^{mut}Rif1^{-/-}$ cells were, as expected, severely impaired in the process, $Brca1^{mut}Rif1^{S \to A}$ cell lines all switched proficiently from expressing IgM to IgA (*Figure 2I*). This finding indicates that phosphorylation of the conserved IDR-CII SQ motifs is dispensable for physiological levels of CSR.

Finally, we assessed whether phosphorylation of the IDR-CII SQ cluster modulates RIF1's role in the regulation of DSB resection following ionizing irradiation (IR)-induced DNA damage. To do so, we compared the phosphorylation levels of replication protein A (RPA) in RIF1-proficient, -deficient, and RIF1$^{S \to A}$-expressing $Brca1^{mut}$ cells. RPA is a heterotrimeric complex (RPA70, RPA32, and RPA14 subunits) that binds to single-stranded DNA (ssDNA) with high affinity (*Maréchal and Zou, 2015*). Defects in DSB end protection lead to hyperphosphorylation of the RPA32 subunit on S4/S8 upon IR exposure (*Maréchal and Zou, 2015*; *Noordermeer et al., 2018*). As expected, IR induced a marked phosphorylation of RPA32 in $Brca1^{mut}Rif1^{-/-}$ cells (*Figure 2—figure supplement 2H*). In contrast, $Brca1^{mut}Rif1^{S \to A}$ cells were as proficient as controls in counteracting RPA32 phosphorylation following IR-induced DSBs (*Figure 2—figure supplement 2H*).

We concluded that RIF1-mediated DSB end protection activity is not dependent on the phosphorylation of the conserved IDR-CII SQ cluster.

## Phosphorylation of the IDR-CII SQ cluster enables RIF1-dependent protection of stalled DNA replication forks

RIF1 has recently been reported to play a genome-protective role under conditions of DNA replication stress (*Ray Chaudhuri et al., 2016*; *Garzón et al., 2019*; *Mukherjee et al., 2019*). RIF1 is recruited to stalled DNA replication forks where it protects nascent DNA from degradation by the DNA2 nuclease in a manner dependent on its interaction with protein phosphatase 1 (PP1) (*Ray Chaudhuri et al., 2016*; *Garzón et al., 2019*; *Mukherjee et al., 2019*). This activity allows for timely restart of stalled forks and prevents genome instability (*Garzón et al., 2019*; *Mukherjee et al., 2019*). Given the high proliferative nature of the cellular context where phosphorylation of the conserved IDR-CII SQ motifs was originally detected (activated primary B cells, *Figure 1*), we asked whether these PTMs could influence RIF1 function during replication stress.

BRCA1 plays a protective role at DNA replication forks that is independent from RIF1 (*Ray Chaudhuri et al., 2016*; *Garzón et al., 2019*; *Mukherjee et al., 2019*; *Schlacher et al., 2012*). Therefore, to specifically address the contribution of RIF1 phosphorylation to fork protection, we first generated a set of *Rif1* knockout and A$^{1387}$A$^{1416}$A$^{1528}$-bearing phosphomutant cell lines on a BRCA1-proficient background (WT CH12 cells) ($Rif1^{-/-}$ and $Rif1^{S \to A}$, *Figure 3A*, *Figure 3—figure supplement 1A–C*). As expected, deletion of RIF1 severely impaired CSR (*Chapman et al., 2013*; *Di Virgilio et al., 2013*; *Escribano-Díaz et al., 2013*; *Figure 3—figure supplement 1D*), whereas, in agreement with the findings from the BRCA1-deficient background (*Figure 2I*), CSR was not affected in $Rif1^{S \to A}$ cell lines (*Figure 3—figure supplement 1D*). Furthermore, analogously to what we described in the $Brca1^{mut}$ setting (*Figure 2—figure supplement 2H*), $Rif1^{S \to A}$ cells did not display the IR-induced RPA phosphorylation that was detectable in the absence of RIF1 (*Figure 3—figure supplement 1E*).

Next, we applied the DNA fiber assay to monitor the degradation of nascent DNA at forks that were stalled via treatment with hydroxyurea (HU) (*Figure 3B*). HU interferes with DNA synthesis by inhibiting ribonucleotide reductase, the rate-limiting enzyme in dNTP synthesis (*Singh and Xu, 2016*). Both $Rif1^{-/-}$ and $Brca1^{mut}$ genotypes exhibited the expected fork degradation phenotype (*Figure 3C*), thus indicating that the protective pathways mediated by these factors are active also in CH12 cells.

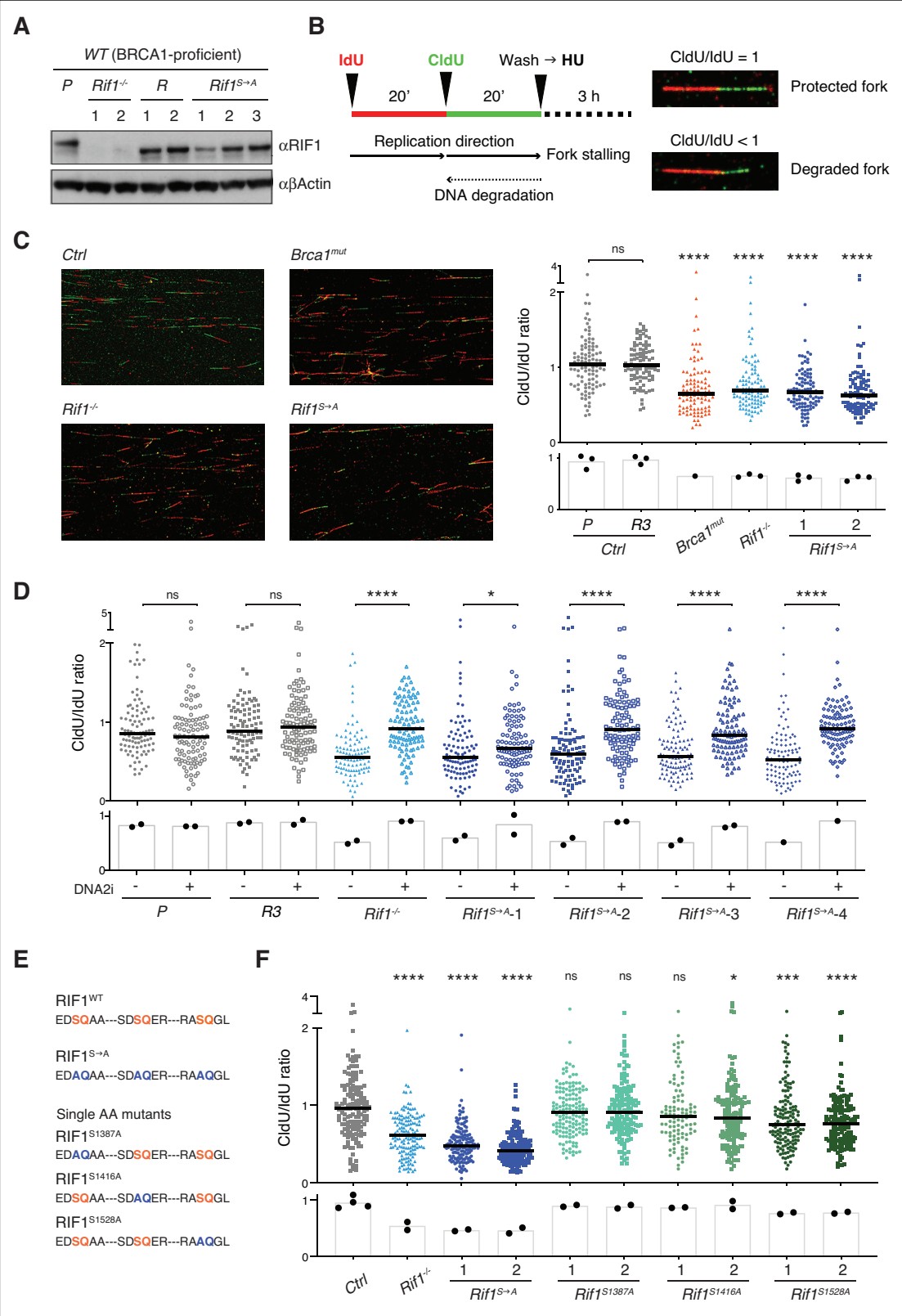

**Figure 3.** Phosphorylation of the conserved IDR-CII serine-glutamine (SQ) cluster enables RIF1-dependent protection of nascent DNA at stalled replication forks. (**A**) Western blot analysis of whole-cell extracts from independent cells lines of the indicated genotypes (*Rif1⁻/⁻*, control Random clones *R*, and *Rif1ˢ→ᴬ*, all generated on the parental – *P* – WT CH12 background). (**B**) Left: schematic representation of the DNA fiber assay employed to assess protection of nascent DNA at stalled replication forks. Right: representative images of protected and degraded DNA fibers. (**C**) Left: representative

*Figure 3 continued on next page*

*Figure 3 continued*

fields for the analysis of nascent DNA degradation following 3 hr treatment with 4 mM HU in CH12 cells of the indicated genotypes. Right: graph summarizing the quantification of CldU/IdU ratio for n = 100 DNA fibers analyzed per genotype (1 and 2 indicate two different $Rif1^{S \to A}$ clonal derivatives). The graph is representative of three independently performed experiments. (**D**). Graph summarizing the quantification of CldU/IdU ratio for n ≥ 100 DNA fibers analyzed per genotype in HU-treated cells in the absence/presence of 0.3 μM DNA2i (four different $Rif1^{S \to A}$ clonal derivatives were employed). The graph is representative of two independently performed experiments. (**E**) Amino acid sequence in the IDR-CII SQ region of WT, S→A- and single SQ-mutated RIF1 proteins. (**F**) Graph summarizing the quantification of CldU/IdU ratio for n = 100–150 DNA fibers analyzed per genotype (1 and 2 indicate two different clonal derivatives). The control (*Ctrl*) samples comprise the parental CH12 cell line and the *R* clone employed also for the analyses in panels (**C**) and (**D**). The graph is representative of two independently performed experiments. Significance in panels (**C**), (**D**), and (**F**) was calculated with the Mann–Whitney *U*-test, and the median is indicated. Significance for each cell line in the graph in panel (**C**) was calculated in reference to the parental CH12 (P) sample. ns, not significant; *p≤0.05; ***p≤0.001; ****p<0.0001. The bar charts underneath the main graphs in panels (**C**), (**D**), and (**F**) display the samples' median for each independently performed experiment.

The online version of this article includes the following source data and figure supplement(s) for figure 3:

**Source data 1.** Original file for the Western blot analysis in *Figure 3A* (anti-RIF1 and anti-β-actin).

**Source data 2.** PDF containing *Figure 3A* and original scans of the relevant Western blot analysis (anti-RIF1 and anti-β-actin) with highlighted bands and sample labels.

**Source data 3.** Original image of control DNA fibers in *Figure 3C*.

**Source data 4.** Original image of $Brca1^{mut}$ DNA fibers in *Figure 3C*.

**Source data 5.** Original image of $Rif1^{-/-}$ DNA fibers in *Figure 3C*.

**Source data 6.** Original image of $Rif1^{S \to A}$ DNA fibers in *Figure 3C*.

**Figure supplement 1.** Generation of RIF1-mutant CH12 cell lines on a WT background.

**Figure supplement 1—source data 1.** Original file for the Western blot analysis in *Figure 3—figure supplement 1B* (anti-RIF1 and anti-β-actin).

**Figure supplement 1—source data 2.** PDF containing *Figure 3—figure supplement 1B* and original scans of the relevant Western blot analysis (anti-RIF1 and anti-β-actin) with highlighted bands and sample labels.

**Figure supplement 1—source data 3.** Original file for the diagnostic digestion of Ctrl samples in *Figure 3—figure supplement 1C*.

**Figure supplement 1—source data 4.** Original file for the diagnostic digestion of $Rif1^{S \to A}$ in *Figure 3—figure supplement 1C*.

**Figure supplement 1—source data 5.** PDF containing *Figure 3—figure supplement 1C* and original images of the relevant diagnostic digestion with highlighted bands and sample labels.

**Figure supplement 1—source data 6.** Original file for the Western blot analysis in *Figure 3—figure supplement 1E* (anti-pRPA and loading control).

**Figure supplement 1—source data 7.** Original file for the Western blot analysis in *Figure 3—figure supplement 1E* (anti-RPA).

**Figure supplement 1—source data 8.** PDF containing *Figure 3—figure supplement 1E* and original scans of the relevant Western blot analysis (anti-pRPA and anti-RPA) with highlighted bands and sample labels.

**Figure supplement 1—source data 9.** Original file for the diagnostic digestion of $Rif1^{S1387A}$ clonal cell lines in *Figure 3—figure supplement 1F*.

**Figure supplement 1—source data 10.** Original file for the diagnostic digestion of $Rif1^{S1416A}$ clonal cell lines in *Figure 3—figure supplement 1F*.

**Figure supplement 1—source data 11.** Original file for the diagnostic digestion of $Rif1^{S1528A}$ clonal cell lines in *Figure 3—figure supplement 1F*.

**Figure supplement 1—source data 12.** PDF containing *Figure 3—figure supplement 1F* and original images of the relevant diagnostic digestion with highlighted bands and sample labels.

**Figure supplement 1—source data 13.** Original file for the Western blot analysis of $Rif1^{S1387A}$ clonal cell lines in *Figure 3—figure supplement 1G* (anti-RIF1 and anti-β-actin).

**Figure supplement 1—source data 14.** Original file for the Western blot analysis of $Rif1^{S1416A}$ clonal cell lines in *Figure 3—figure supplement 1G* (anti-RIF1 and anti-β-actin).

**Figure supplement 1—source data 15.** Original file for the Western blot analysis of $Rif1^{S1528A}$-1 clonal cell line in *Figure 3—figure supplement 1G* (anti-RIF1 and anti-β-actin).

**Figure supplement 1—source data 16.** Original file for the Western blot analysis of $Rif1^{S1528A}$-2 clonal cell line in *Figure 3—figure supplement 1G* (anti-RIF1).

**Figure supplement 1—source data 17.** Original file for the Western blot analysis of $Rif1^{S1528A}$-2 clonal cell line in *Figure 3—figure supplement 1G* (anti-β-actin).

**Figure supplement 1—source data 18.** PDF containing *Figure 3—figure supplement 1G* and original scans of the relevant Western blot analysis (anti-RIF1 and anti-β-actin) with highlighted bands and sample labels.

Interestingly, $Rif1^{S \to A}$ clonal derivatives showed increased degradation of stalled forks compared to controls and to the same levels observed in $Rif1^{-/-}$ cells (*Figure 3C*), thus suggesting that abrogation of these IDR-CII SQ phosphorylation events prevents RIF1 function at the forks.

RIF1 protective role at stalled forks is dependent on the ability of its interactor PP1 to dephosphorylate and inactivate DNA2, which in turn limits the nuclease-mediated processing of DNA replication forks (*Garzón et al., 2019*; *Mukherjee et al., 2019*). To confirm the DNA2 dependency of the fork degradation phenotype observed in cells expressing phosphomutant RIF1 protein, we repeated the DNA fiber assay in the presence of the DNA2 inhibitor NSC-105808 (DNA2i) (*Garzón et al., 2019*; *Kumar et al., 2017*). Analogously to the result observed in $Rif1^{-/-}$ cells (*Figure 3D*; *Garzón et al., 2019*; *Mukherjee et al., 2019*), DNA2i treatment rescued the fork degradation phenotype in all HU-treated $Rif1^{S \to A}$ clonal derivatives (*Figure 3D*).

We next asked whether the fork degradation phenotype exhibited by $Rif1^{S \to A}$ cells was mediated by the abrogation of phosphorylation at a specific SQ site within the IDR-CII SQ cluster. To answer this question, we generated single SQ mutant CH12 cell lines and assessed their capability to protect nascent DNA at stalled forks via the DNA fiber assay (*Figure 3E*, *Figure 3—figure supplement 1F and G*). We found that while $Rif1^{S1387A}$ and $Rif1^{S1416A}$ cell lines were proficient in protecting stalled forks from degradation, $Rif1^{S1528A}$ cells exhibited a reproducible fork degradation defect (*Figure 3F*). However, the phenotype was modest and did not recapitulate the severe defect of $Rif1^{-/-}$ and $Rif1^{S \to A}$ cells (*Figure 3F*). Altogether, this data suggests that phosphorylation of S1528 contributes to, but is not sufficient for, fork protection, and that multiple phosphorylation events within the IDR-CII SQ cluster are responsible for RIF1's ability to protect nascent DNA under conditions of replication stress.

We concluded that phosphorylation of the conserved IDR-CII SQ cluster enables RIF1-dependent inhibition of DNA2 activity and protection of nascent DNA at stalled replication forks.

## Phosphorylation of the IDR-CII SQ cluster promotes HU-induced recruitment of RIF1 to DNA replication forks

To mechanistically dissect how phosphorylation of the conserved IDR cluster contributes to RIF1's role in protection of stalled DNA replication forks, we first assessed the integrity of RIF1-PP1 interaction via co-immunoprecipitation studies. We found that RIF1$^{S \to A}$ mutant protein retains the ability to interact with PP1, thus indicating that the abrogation of phosphorylation events in the conserved cluster does not have a major impact on RIF1-PP1 association (*Figure 4A*).

Next, we asked whether phosphorylation of IDR-CII SQ influences RIF1 recruitment to stalled DNA replication forks. To do so, we applied a proximity ligation assay (PLA) that employs flow cytometry measurements to quantitatively assess the localization of RIF1 at sites of EdU incorporation in the presence and absence of HU. As expected (*Garzón et al., 2019*; *Mukherjee et al., 2019*), RIF1 and EdU co-localization increased upon HU treatment in control cell lines (*Figure 4B*). In contrast, the HU-induced RIF1-EdU proximity signal was only modestly affected in $Rif1^{S \to A}$ clonal derivatives (*Figure 4B*). This data suggests that phosphorylation of the IDR-CII SQ cluster facilitates RIF1 interaction with stalled replication forks.

Finally, we investigated the dependency of phosphorylation events within the IDR-CII SQ cluster on replication stress. To this end, we have optimized RIF1 pull-downs for the identification of phosphosites in primary B cells and compared the RIF1 peptide composition of mock- versus HU-treated samples in the absence and presence of ATM or ATR inhibitors (*Figure 4C*). Interestingly, the only SQ site that was identified to be phosphorylated in an HU-dependent manner was indeed one of the three conserved motifs of the IDR-CII SQ cluster, S1416 (*Figure 4D*). In addition, HU-induced phosphorylation of S1416 was reduced following treatment with ATM and, to a lesser extent ATR, inhibitors (*Figure 4D*). Although this new dataset did not yield additional phosphorylated SQ motifs, we cannot exclude the likely possibility that peptides containing phospho-S1387 and phospho-S1528 residues might simply be undetectable under the conditions employed for this new set of pull-downs and mass spectrometry analysis. In support of this point, an independent proteomics analysis of hRIF1 isolated from Flp-In T-REx GFP-RIF1-L cells (*Watts et al., 2020*) identified S1542 (which corresponds to S1528 in mouse RIF1, see *Figure 1—source data 2*) as an SQ site phosphorylated following treatment with the DNA polymerase inhibitor aphidicolin, which also induces replication stress (*Figure 4E*). Collectively, these results build on the initial identification of S1416 and S1528 in the RIF1 I-DIRT

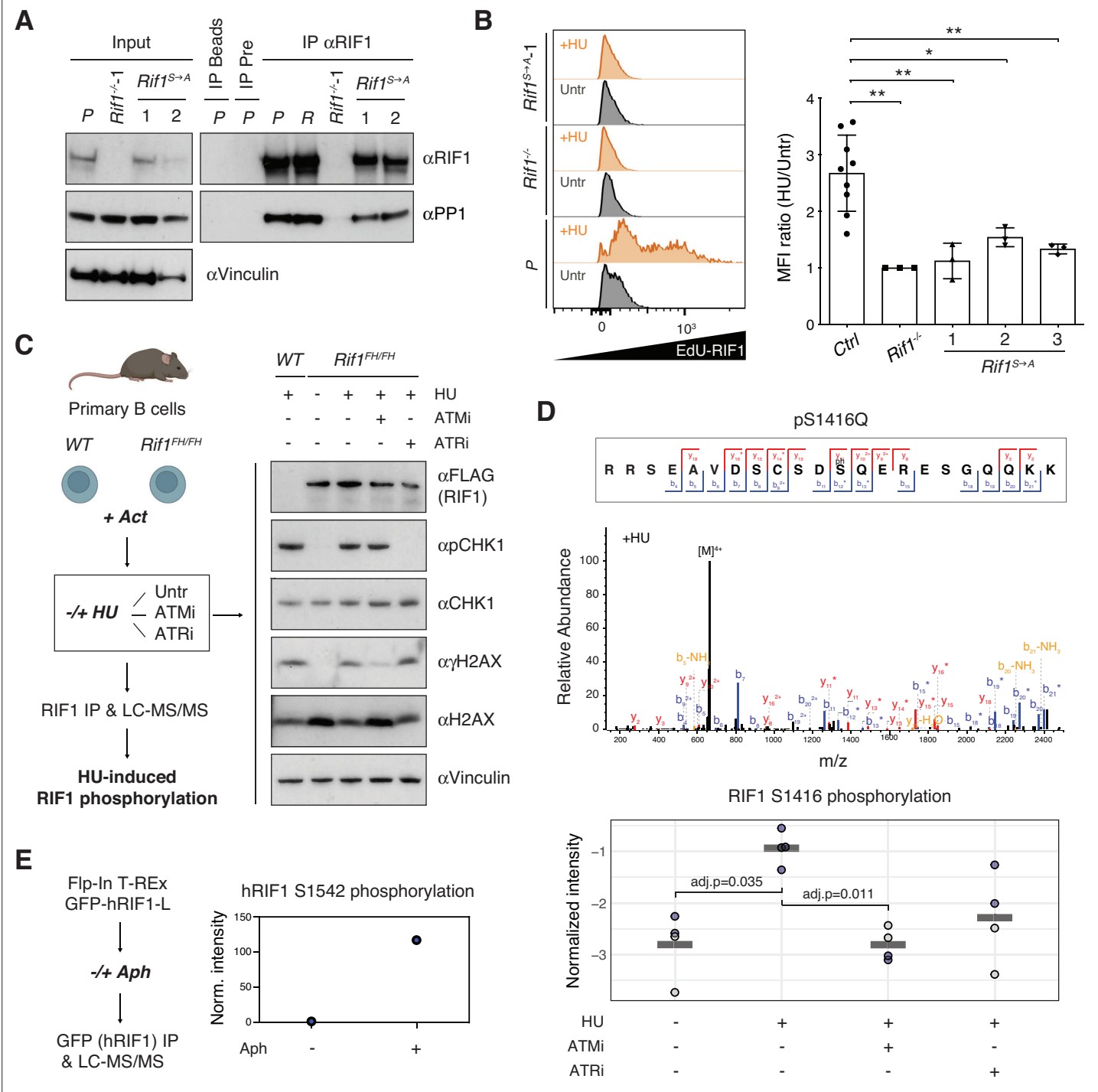

**Figure 4.** Phosphorylation of the IDR-CII serine-glutamine (SQ) cluster promotes hydroxyurea (HU)-induced recruitment of RIF1 to replication forks. (**A**) Western blot analysis of anti-RIF1 immunoprecipitates (IP) from cell lines of the indicated genotypes (WT parental CH12 cells [P], a validated Random clone [R], *Rif1^-/-^*, and two different *Rif1^S→A^* clonal derivatives). The analysis is representative of two independently performed experiments. Pre: pre-immune serum control for αRIF1 IP. (**B**) Left: representative histograms displaying EdU-RIF1 proximity signal in untreated (Untr) and HU-treated (+HU) samples of the indicated genotypes. Right: summary graph showing quantification of the proximity signal data for three independent experiments. For each sample, values were expressed as fold mean fluorescent intensity (MFI) of HU-treated over untreated conditions, and normalized within each experiment to *Rif1^-/-^*, which was set to 1. Samples include parental CH12 cells and two validated Random clones as positive (*Ctrl*), and *Rif1^-/-^* cells as negative, experimental controls, and three different *Rif1^S→A^* clonal derivatives. (**C**) Left: schematic representation of the strategy for the identification of HU-induced RIF1 phosphosites in primary B cells. Act: activation; Untr: untreated (no ATMi/ATRi); LC-MS/MS: liquid chromatography-tandem mass

*Figure 4 continued on next page*

*Figure 4 continued*

spectrometry. Right: representative Western blot analysis of whole-cell extracts employed for the RIF1 pull-downs. The analysis is representative of the four mice pairs (*WT* and *Rif1^FH/FH^*) included in the mass spec experiment. (**D**) Top: representative annotated MS/MS spectra of a RIF1 peptide encompassing phosphorylated $S^{1416}$ residue from one HU-treated *Rif1^FH/FH^* sample. Bottom: graph summarizing $S^{1416}$ phosphosite intensities in the different conditions shown in panel (**C**). Values were normalized to bait protein (RIF1) levels for each sample followed by replicate-wise median normalization. The horizontal line indicates the mean of the four data points. Adjusted p-values shown were calculated using a Benjamini–Hochberg correction after a global two-sample moderated *t*-test. Values for *t*-test were imputed using a Gaussian distribution with downshift by column after filtering for at least 60% valid values per row across all samples (without WT). Original values are shown in blue, imputed values in gray. (**E**) Left: schematic representation of the strategy for the identification of aphidicolin-induced hRIF1 phosphosites. GFP-hRIF1-L: human RIF1 long isoform fused to GFP; Aph: aphidicolin. Right: hRIF1 $S^{1542}$ phosphosite intensity values were normalized to bait protein (hRIF1) levels and shown as fold increase of Aph- versus DMSO-treated sample, which was set to 1. Norm: normalized. Significance in panel (**B**) was calculated with the Mann–Whitney *U*-test, and error bars represent SD. *p≤0.05; **p≤0.01.

The online version of this article includes the following source data for figure 4:

**Source data 1.** Original file for the Western blot analysis of input and immunoprecipitate (IP) in *Figure 4A* (anti-RIF1).

**Source data 2.** Original file for the Western blot analysis of input in *Figure 4A* (anti-PP1).

**Source data 3.** Original file for the western blot analysis of IP in *Figure 4A* (anti-PP1).

**Source data 4.** Original file for the Western blot analysis of input in *Figure 4A* (anti-vinculin).

**Source data 5.** PDF containing *Figure 4A* and original scans of the relevant Western blot analysis (anti-RIF1, anti-PP1, and anti-vinculin) with highlighted bands and sample labels.

**Source data 6.** Original file for the Western blot analysis in *Figure 4C* (anti-FLAG).

**Source data 7.** Original file for the Western blot analysis in *Figure 4C* (anti-pCHK1).

**Source data 8.** Original file for the western blot analysis in *Figure 4C* (anti-CHK1 and anti-H2AX).

**Source data 9.** Original file for the Western blot analysis in *Figure 4C* (anti-vinculin and anti-γH2AX).

**Source data 10.** PDF containing *Figure 4C* and original scans of the relevant Western blot analysis (anti-FLAG, anti-pCHK1, anti-CHK1, anti-H2AX, anti-vinculin, and anti-γH2AX) with highlighted bands and sample labels.

preparations (*Figure 1E and F*) and demonstrate that SQ motifs in the conserved cluster are phosphorylated following treatment with replication stress-causing agents.

Altogether, these findings suggest that replication stress induces phosphorylation events within the IDR-CII SQ cluster that promote RIF1 recruitment to stalled replication forks and protection of nascent DNA. The key players and precise molecular mechanism underlying phosphorylation-dependent recruitment of RIF1 to replicated DNA will be the object of future studies.

## Discussion

The regulation of DNA processing and its consequences for the preservation of genome integrity have important clinical implications. On the one hand, downregulation or inactivating mutations in DSB end protection factors confer resistance to PARP inhibitors in BRCA1-deficient tumors in mice and a patient-derived model (*Dev et al., 2018*; *Jaspers et al., 2013*; *Noordermeer et al., 2018*; *Xu et al., 2015*). On the other hand, protection of DNA replication forks has recently been proposed as a mechanism for chemoresistance in the context of BRCA deficiency (*Ray Chaudhuri et al., 2016*). Hence, the dissection of pathways and molecular determinants in the regulation of DNA processing has profound implications for the development and improvement of targeted antitumoral treatments.

RIF1 plays at least two, and to some extent conflicting, roles in the preservation of genome integrity during DNA replication: a genome-protective role in stabilizing nascent DNA at stalled but unbroken forks, and a potentially genome-destabilizing role in regulating DNA repair by opposing resection at DSBs. Both roles depend on RIF1's ability to control DNA processing, albeit on different DNA substrates and via independent mechanisms: the protection of newly replicated DNA at stalled forks through PP1-induced DNA2 inactivation (*Garzón et al., 2019*; *Mukherjee et al., 2019*), and the inhibition of DSB resection at collapsed forks via the 53BP1-triggered cascade (*Chapman et al., 2013*; *Escribano-Díaz et al., 2013*; *Feng et al., 2013*; *Zimmermann et al., 2013*), respectively. Given the impact of these pathways on genome stability and cell viability, it is likely that multiple layers of regulations have evolved to ensure the coordination of RIF1 activities in the control of DNA processing. In this study, we identified three serine residues that are phosphorylated in hyperproliferative B

lymphocytes. This set of phosphosites is specifically required for the role of RIF1 at stalled forks, and as such, exerts a genome protective function under conditions of DNA replication stress.

Interestingly, the identified PTMs occur within a cluster of conserved SQ motifs in the IDR of mammalian RIF1. IDRs are stretches of sequences that do not adopt any stable, defined secondary or tertiary structures (*Wright and Dyson, 2015*). Proteins characterized by a high degree of intrinsic disorder rapidly transition between different folding states. Phosphorylation of key residues within IDRs has been shown to influence protein folding, interaction with binding partners, and, as a consequence, protein function in several biological settings (*Bah and Forman-Kay, 2016*; *Wright and Dyson, 2015*). As a relevant example, phosphorylation of 53BP1 SQ/TQ motifs within its intrinsically disordered N-terminus is essential for the DNA damage-dependent recruitment of RIF1 to sites of damage and protection against DSB resection (*Chapman et al., 2013*; *Di Virgilio et al., 2013*; *Escribano-Díaz et al., 2013*; *Feng et al., 2013*; *Silverman et al., 2004*; *Zimmermann et al., 2013*). In this study, we showed that abrogation of replication stress-induced phosphorylation of a cluster of conserved SQ motifs in RIF1 IDR impairs its recruitment to stalled DNA replication forks and results in DNA2-mediated degradation of nascent DNA.

Orthologous IDRs exhibit molecular features that are crucial for function but do not translate into any noticeable similarity at the level of primary amino acid sequences (*Zarin et al., 2019*; *Zarin et al., 2017*). These molecular features, which include, for instance, length, complexity, and net charge, appear to be under evolutionary selection, thus explaining how the functional output of IDRs could be maintained despite highly divergent amino acid sequences (*Zarin et al., 2019*; *Zarin et al., 2017*). The phosphorylation of a set of IDR SQ motifs that we report in this study for mammalian RIF1 could represent such an evolutionary signature. In this regard, the IDR of *Saccharomyces cerevisiae* Rif1 contains a cluster of seven SQ/TQ consensus motifs for the ATM/ATR yeast homologs Tel1/Mec1, some of which have been reported to be phosphorylated in vivo (*Smolka et al., 2007*; *Sridhar et al., 2014*; *Wang et al., 2018*). Interestingly, a recent bioRxiv manuscript showed that abrogation of phosphorylation at these seven SQ/TQ sites in yeast Rif1 impaired DNA replication fork protection after treatment with HU (*Monerawela et al., 2020*). Although RIF1 IDRs exhibit low conservation across evolution, the identification of a cluster of SQ/TQ motifs whose phosphorylation influences fork protection in both mammalian and yeast RIF1 hints at an evolutionary conserved mechanism, and underlying molecular feature, for the regulation of nascent DNA processing under conditions of DNA replication stress.

## Materials and methods
### Mice and derived primary cell cultures

*Rif1^FH/FH^* (*Cornacchia et al., 2012*) and *Rif1^F/F^Cd19^Cre/+^* (*Di Virgilio et al., 2013*) mice were previously described and maintained on a C57BL/6 background. Mice were kept in a specific pathogen-free (SPF) barrier facility under standardized conditions (20 ± 2°C temperature; 55% ± 15% humidity) on a 12 hr light/12 hr dark cycle. Animals were maintained in small groups (4–5) or as breeding pairs in individually ventilated cages to ensure optimal habitat condition. Mice of both genders were used for the experiments. All experiments were performed in compliance with the European Union (EU) directive 2010/63/EU, and in agreement with Landesamt für Gesundheit und Soziales directives (LAGeSo, Berlin, Germany).

Primary cell cultures of resting B lymphocytes were isolated from *WT, Rif1^FH/FH^*, and *Rif1^F/F^Cd19^Cre/+^* mouse spleens using anti-CD43 MicroBeads (Miltenyi Biotec), and grown in RPMI 1640 medium (Life Technologies) supplemented with 10% fetal bovine serum (FBS), 10 mM HEPES (Life Technologies), 1 mM sodium pyruvate (Life Technologies), 1× Antibiotic Antimycotic (Life Technologies), 2 mM L-glutamine (Life Technologies), and 1× 2-mercaptoethanol (Life Technologies) at 37°C and 5% $CO_2$ levels. Naïve B cells were activated by addition of 25 µg/ml LPS (Sigma-Aldrich), 5 ng/ml of mouse recombinant IL-4 (Sigma-Aldrich), and 0.5 µg/ml anti-CD180 (RP/14) (BD Biosciences) to the cultures upon isolation.

Primary MEFs (pMEFs) were isolated from *WT* and *Rif1^FH/FH^* mice as follows. Pregnant mice were sacrificed on day E12.5 by cervical dislocation, and embryos were removed from uterine horns and placed individually in plates containing PBS (Thermo Fisher Scientific). Brain, tail, limbs, and dark red organs were removed and the remaining tissue was transferred into fresh PBS. Tissue was treated with 2 ml of Trypsin-EDTA 0.05% (Gibco) at 37°C for 15 min, and cell suspension was passed through

a syringe with 18-gauge needle. Trypsin was neutralized with DMEM medium (Life Technologies) supplemented with 10% FBS, 2 mM L-glutamine, and Penicillin-Streptomycin (Life Technologies). pMEFs from each embryo were expanded in 25 cm plates at 37°C and 5% $CO_2$ levels to reach 80% confluency, and either used immediately for immortalization (see below) or frozen.

## Cell lines

The cell lines employed for this study are CH12 (CH12F3, mouse, *Nakamura et al., 1996*); *Rif1$^{-/-}$* CH12 (clone 1, mouse, *Delgado-Benito et al., 2018*); WT (Random clones), *Rif1$^{-/-}$* (clone 2), and *Brca1$^{mut}$* CH12 clonal derivatives (mouse, this paper), as well as RIF1 phosphomutant CH12 cell lines generated on both WT and *Brca1$^{mut}$* backgrounds (mouse, this paper); *WT* and *Rif1$^{FH/FH}$* immortalized mouse embryonic fibroblasts (iMEFs, this paper). iMEFs were generated by immortalization of the pMEFs cultures described above via retroviral transduction of a construct expressing the SV40 T-antigen.

CH12 cells were grown in RPMI 1640 medium supplemented with 10% FBS, 10 mM HEPES, 1 mM sodium pyruvate, 1× Antibiotic Antimycotic, 2 mM L-glutamine, and 1× 2-mercaptoethanol at 37°C and 5% $CO_2$ levels. iMEFs were cultured in DMEM medium supplemented with 10% FBS, 2 mM L-glutamine, and Penicillin-Streptomycin at 37°C and 5% $CO_2$ levels.

Mycoplasma contamination was not detected in any cell line tested in the lab using commercially available mycoplasma detection kits.

## Identification of RIF1 phosphoresidues

RIF1 phosphoresidues were identified via analysis of RIF1 I-DIRT samples (*Figure 1E and F*) prepared from primary B cell cultures as previously described (*Delgado-Benito et al., 2018*), with the only difference that preparations with varying concentrations of glutaraldehyde (1–5 mM) were employed. Samples were loaded on NuPAGE Bis-Tris Gels (Thermo Fisher Scientific) and run for a short time to produce gel plugs. The gel samples were subjected to in-gel tryptic digestions. Peptides were extracted, purified, and analyzed by LC-MS using a Thermo Orbitrap Fusion mass spectrometer, with a Thermo Easy-nLC 1000 HPLC and a Thermo Easy-Spray electrospray source. Isotopically labeled proteins were identified by searching against a mouse protein sequence database using the GPM software (*Beavis, 2006*), which was set to search for tryptic peptides whose lysines and arginines were isotopically labeled and for potential phosphorylation modifications at serines, threonines, and tyrosines.

HU-induced phosphorylation of RIF1 in mouse B lymphocytes (*Figure 4C and D*) was performed as follows. Splenocytes isolated from *WT* and *Rif1$^{FH/FH}$* mice were treated at 72 hr post-activation with either 25 nM ATRi (BAY 1895344, Selleckchem), 20 nM ATMi (AZD0156, Selleckchem), or DMSO control for 20 min, followed by 4 mM HU (Sigma-Aldrich) or mock control for 3 hr. Cells were harvested, washed twice with ice-cold 1× PBS, and snap-frozen in liquid nitrogen. Cells were lysed at 4°C for 10 min in lysis buffer (150 mM NaCl, 50 mM Tris-HCl, 1% IGEPAL CA-630 [Sigma-Aldrich], 5% glycerol, 0.5% deoxycholate, and 0.1% sodium dodecyl sulfate) supplemented with MS-SAFE Protease and Phosphatase Inhibitor (Sigma-Aldrich), 5 mM sodium butyrate (HDACs inhibitor, Sigma-Aldrich), 5 mM 2-chloroacetamide (deubiquitinase inhibitor, Sigma-Aldrich), and Benzonase (Sigma-Aldrich). Lysates were clarified at 14,000 rpm for 10 min at 4°C and immediately used for the immunoprecipitation reaction. Protein G Dynabeads (Thermo Fisher) were conjugated with anti-HA antibody (Santa-Cruz, 2 µg/mg of whole-cell extracts) for 1 hr at room temperature (RT). Conjugated beads were washed two times with lysis buffer and incubated with lysates at 4°C for 1 hr. Beads were washed two times with wash buffer (150 mM NaCl, 50 mM Tris-HCl, 0.05% IGEPAL CA-630, 5% glycerol) and two times with wash buffer without IGEPAL CA-630, and snap-frozen in liquid nitrogen.

Tryptic on-bead digestion was carried out following essentially the protocol from *Hubner et al., 2010*. Proteins were digested from the beads in the presence of 2 M urea, 50 mM Tris pH 7.5, 1 mM dithiothreitol (DTT), and 5 µg/ml trypsin (Promega) at 25°C for 1 hr. Eluted bead-free pre-digested material was reduced with 4 mM DTT at 25°C for 30 min followed by an alkylation step with 10 mM iodoacetamide at 25°C for 30 min. Main digest occurred by addition of 1 µg trypsin at 25°C overnight. Samples were acidified by adding 1% (v/v) formic acid and then desalted on stage tips (*Rappsilber et al., 2007*). Eluted peptides were subjected to a modified SP3 procedure for an additional cleanup on peptide level (*Hughes et al., 2019*). Specifically, peptides were precipitated on 1 mg SP3 bead mix (Sera-Mag A and Sera-Mag B beads, GE Healthcare) by adding acetonitrile to a final concentration of

≥98% (v/v). After incubation for 20 min and three washes with pure acetonitrile, samples were eluted twice with 50 µl water. After lyophilization, samples were dissolved in MS sample buffer (3% [v/v] acetonitrile, 0.1% [v/v] formic acid). LC-MS measurements were carried out on an orbitrap Exploris 480 mass spectrometer (Thermo Fisher Scientific) coupled to an EASY-nLC 1200 system (Thermo Fisher Scientific) applying a 110 min gradient in data-dependent MS2-mode. MS1 resolution was set to 60,000 for a scan range of 300–1800, MS2 resolution was specified to 15,000 while the maximum injection time for MS2 was set to 100 ms with an isolation width of 1.3 *m/z*.

Analysis was done in MaxQuant (version 1.5.2.8; *Cox and Mann, 2008*) applying an Andromeda search against a UniProt mouse database (2018) plus common contaminants and a false discovery rate of 0.01 on peptide as well as site level while using the match-between-runs feature. RIF1 was identified with an overall sequence coverage of 65.6%. Phosphorylation on serine, threonine and tyrosine, acetylation on protein N-termini, as well as oxidized methionine were set as variable modifications. Carbamidomethylation on cysteine was set as fixed modification. The number of maximum missed cleavages was set to 5, and the number of allowed variable modifications specified to 4. Phosphosite intensities were normalized to the bait protein. A requirement of at least 60% valid values (across all samples except the WT control) was used to filter for phospho-STY sites for quantitation and further normalized by median. For two-sample moderated *t*-testing (limma R package; *Ritchie et al., 2015*) across all sites, imputation was applied by replicate using a randomized Gaussian distribution with a width of 0.2 and a downshift of 1.8. Significance calling on sites was done after multiple comparison correction by calculating adjusted p-values with the Benjamini–Hochberg method.

Analysis of aphidicolin-induced phosphorylation of hRIF1 (*Figure 4E*) was performed using Flp-In T-REx GFP-RIF1-L cells (HEK293-derived Flp-In T-REx 293 cells expressing GFP-hRIF1-L, *Watts et al., 2020*) as follows. Flp-In T-REx GFP-RIF1-L cells were cultivated in DMEM medium and induced for GFP-RIF1-L expression by addition of 1 µg/ml doxycycline (Sigma-Aldrich) 48 hr before harvesting. Cells were treated with 1 µM aphidicolin (Abcam) or DMSO (for mock control) for the final 24 hr. Cells were gently washed in dishes with 1× ice-cold Tris-buffered saline (TBS), lysed, and gently scraped off in ice-cold TBS IP buffer (1× TBS supplemented with 1% CHAPS, 1× Halt protease and phosphatase inhibitor cocktail [Thermo Fisher], and 1 mM phenylmethylsulfonyl fluoride [PMSF]) supplemented with 3 mM MgCl$_2$ and Benzonase. Lysates were incubated for 30 min at 4°C with gentle agitation. The lysate was spun at 20,000 for 10 min, and the supernatant then used for immunoprecipitation using GFP-Trap Magnetic-agarose beads (Chromotek). Immunoprecipitation was carried out according to the manufacturer's instruction but in Tris IP buffer. Beads were further washed with 100 mM ammonium bicarbonate and on-beads trypsin digestion was performed essentially as described (*Garzón et al., 2019*) but without Cys alkylation. Peptides were analyzed using an Orbitrap Q Exactive Plus mass spectrometer equipped with nano-LC C18 liquid chromatography over 60 min elution gradient. The raw MS datasets were analyzed using MaxQuant software (version 1.6.2.3). MS1 intensity of each phosphorylation site was normalized by iBAQ value of RIF1 in each sample. The normalized phospho/RIF1 values between samples were compared.

## CSR assay

CH12 cells were stimulated to undergo CSR to IgA by treatment with 1–5 µg/ml αCD40 (BioLegend), 5 ng/ml TGFβ (R&D Systems), and 5 ng/ml of mouse recombinant IL-4 for 48 hr. For class switching analysis, cell suspensions were stained with fluorochrome-conjugated anti-IgA (Southern Biotech) and samples were acquired on an LSRFortessa cell analyzer (BD Biosciences).

## CRISPR-Cas9 gene targeting and generation of CH12 clonal cell lines

Targeting of *Brca1* and *Rif1* loci for generation of indel-bearing clonal derivatives was performed with two gRNA pairs per gene (g*Gene*-N1a and g*Gene*-N1b for Nickase 1, and g*Gene*-N2a and g*Gene*-N2b for Nickase 2) cloned into tandem U6 cassettes in a version of pX330 plasmid (pX330-U6-Chimeric_BB-CBh-hSpCas9, Addgene #42230) mutated to express Cas9$^{D10A}$-T2A-GFP (Nickase-1/2). The Nickase-1/2 constructs were individually transfected into CH12 via electroporation with Neon Transfection System (Thermo Fisher Scientific).

For the generation of *Rif1$^{S→A}$*, *Rif1$^{S1387A}$*, *Rif1$^{S1416A}$*, *Rif1$^{S1528A}$*, and *Brca1$^{mut}$ Rif1$^{S→A}$* cell lines, gRNAs targeting *Rif1* exon 30 (g*RNA*-5' and g*RNA*-3') were cloned into tandem U6 cassettes in a variant of the original pX330 plasmid (pX330-U6-Chimeric_BB-CBh-hSpCas9, Addgene #42230) modified to

express Cas9$^{WT}$-T2A-GFP (pX330-Cas9$^{WT}$-T2A-GFP, kind gift from Van Trung Chu, MDC). CH12 cells (parental WT and *Brca1$^{mut}$*-1) were co-electroporated with the g*RNA*-5′/3′-expressing construct and a circular donor plasmid carrying the synthesized knock-in template (GeneArt Invitrogen). The template was purchased containing all three phosphosites mutated to alanines (A1387, A1416, and A1528) and used for the generation of *Rif1$^{S→A}$* and *Brca1$^{mut}$ Rif1$^{S→A}$* cluster mutant cell lines. For the generation of *Rif1$^{S1387A}$*, *Rif1$^{S1416A}$*, and *Rif1$^{S1528A}$* cell lines, the donor plasmids carrying the individual SQ mutations were individually produced via two rounds of site-directed mutagenesis starting from the original synthesized knock-in template to eventually revert the other two AQ sites back to SQ motifs.

For the generation of both indel- and knock-in-bearing clonal derivatives, single GFP-positive cells were sorted in 96-well plates 40 hr after electroporation and allowed to grow for ca. 12 days before expansion of selected clones. Clonal cell lines were validated at the level of genomic scar (all clonal derivatives), protein level (RIF1 in *Brca1$^{mut}$Rif1$^{-/-}$*, *Brca1$^{mut}$ Rif1$^{S→A}$*, *Rif1$^{-/-}$*, *Rif1$^{S→A}$*, *Rif1$^{S1387A}$*, *Rif1$^{S1416A}$*, and *Rif1$^{S1528A}$*), and phenotypic consequences (BRCA1-deficiency-driven genome instability and lethality in *Brca1$^{mut}$*). Random control cell lines were generated with gRNAs against random sequences not present in the mouse genome (random gRNAs pairs-Cas9$^{D10A}$ constructs).

For in-bulk targeting of *Brca1$^{mut}$*-1 cells in the rescue-of-viability assay, gRNAs against random sequences, *53bp1*, *Rif1*, and *Rev7* genes were cloned into the U6 cassette of pX330-Cas9$^{WT}$-T2A-GFP. *Brca1$^{mut}$*-1 cells were transfected with the Cas9-gRNAs expressing constructs via electroporation, sorted for GFP-positive cells after 40 hr, left to recover for 72 hr, and then treated with 1 μM PARPi for 72 hr before assessment of cell viability.

The sequences of the gRNAs, genotyping, and mutagenesis primers employed in this study are listed in *Table 1*.

## Western blot and co-immunoprecipitation analyses

Western blot analysis of protein levels was performed on whole-cell lysates prepared by lysis in RIPA buffer (Sigma-Aldrich) supplemented with 1 mM DTT (Sigma-Aldrich), cOmplete EDTA-free Protease Inhibitor Cocktail (Roche), and Pierce Phosphatase Inhibitor Mini Tablets (Thermo Fisher). For assessment of RPA phosphorylation, CH12 were seeded at a density of $10^5$ cells/ml and irradiated 24 hr later with 25 Gy, followed by 3 h of recovery time.

For RIF1-PP1 co-immunoprecipitation analysis, exponentially growing CH12 cells were treated with 4 mM HU (Sigma-Aldrich) for 3 hr. Cells were harvested, washed twice with ice-cold 1× PBS, and snap-frozen in liquid nitrogen. Cells were lysed at 4°C for 10 min in lysis buffer (150 mM NaCl, 20 mM Tris-HCl, 0.5% IGEPAL CA-630, 1.5 mM MgCl$_2$ [Sigma-Aldrich]) supplemented with EDTA-free Protease Inhibitor Cocktail, Pepstatin A (Sigma-Aldrich), PMSF (Sigma-Aldrich), phosphatase inhibitors (PhosSTOP, Roche/Sigma-Aldrich), and Benzonase. Lysates were clarified at 14,000 rpm for 10 min at 4°C and immediately used for the immunoprecipitation reactions. Protein A Dynabeads (Thermo Fisher) were conjugated with either anti-RIF1 antibody (4 μg/mg of whole-cell extracts for anti-RIF IP) or equal volume of pre-immune serum (Pre IP control) for 1 hr at RT. Conjugated beads were washed three times with lysis buffer and incubated with lysates at 4°C for 1 hr. Beads were washed five times with lysis buffer, and proteins were eluted by incubation at 72°C for 10 min in NuPAGE LDS sample buffer supplemented with 45 mM DTT.

The antibodies used for co-IP and WB analysis are anti-FLAG M2 (Sigma-Aldrich), FLAG-M2 peroxidase (HRP conjugated, Sigma-Aldrich), anti-HA (Santa-Cruz), pre-immune serum and anti-RIF1 (*Di Virgilio et al., 2013*), anti-PP1 (PPP1A/PPP1CA, Abcam), anti-phospho-RPA32 (S4/S8) (Bethyl Laboratories), anti-RPA32 (Millipore), anti-phospho-CHK1 (S345) (Cell Signaling), anti-CHK1 (Cell Signaling), anti-γH2AX (S139) (Cell Signaling), anti-H2AX (Novus Biologicals), anti-tubulin (Abcam), anti-β-actin (Sigma-Aldrich), and anti-vinculin (Sigma-Aldrich).

## Cell viability and metaphase analysis

For assessment of cell viability, CH12 cells were either mock-treated (DMSO, Carl Roth) or incubated with 1 μM PARPi (Olaparib – AZD2281, Selleckchem) for 72 hr. Residual viability was expressed as percentage of cell viability of PARPi- over DMSO-treated cultures.

For genomic instability analysis, exponentially growing cells were treated with DMSO or 1 μM PARPi for 24 hr followed by 45 min incubation at 37°C with Colcemid (Roche). Metaphase preparation and aberration analysis were performed as follows. Cell pellets were resuspended in 0.075 M KCl at

**Table 1.** List of oligonucleotides used in this study.

| | gRNAs | Sequence (5'→3') | References |
|---|---|---|---|
| | gRandom-1a | GCGAGGTATTCGGCTCCGCG | *Delgado-Benito et al., 2018* |
| | gRandom-1b | ATGTTGCAGTTCGGCTCGAT | *Delgado-Benito et al., 2018* |
| | gBrca1-N1a | GAGCTACCACCGATGTTCCT | This paper |
| | gBrca1-N1b | TCTCAGGGCACAGACTTTGC | This paper |
| | gBrca1-N2a | GCGTTCAGAAAGTTAATGAG | This paper |
| | gBrca1-N2b | TGTTATCCAAGGAACATCGG | This paper |
| | gRif1-N1a | GAAGACCCCTCGGTGCCTCC | *Delgado-Benito et al., 2018* |
| | gRif1-N1b | AAGTCTCCAGAAGCGGCTCC | *Delgado-Benito et al., 2018* |
| | gRif1-N2a | TGTGTGTACCAGGGCACTGT | This paper |
| | gRif1-N2b | ACTCTTAATGATACCATTCA | This paper |
| | gRNA-5' | AAACACTCCGACGGTCTTCG | This paper |
| CRISPR-Cas9 gene targeting for clonal derivative generation | gRNA-3' | CGACTTGTCTAGATTGTCCA | This paper |
| | gRNAs | | |
| | gRandom-1a (as above) | GCGAGGTATTCGGCTCCGCG | *Delgado-Benito et al., 2018* |
| | gRandom-1b (as above) | ATGTTGCAGTTCGGCTCGAT | *Delgado-Benito et al., 2018* |
| | gRandom-1c | GCTTTCACGGAGGTTCGACG | This paper |
| | g53bp1-1 | CAGATGTTTATTATGTGGAT | *Delgado-Benito et al., 2018* |
| | g53bp1-2 | GAGTGTACGGACTTCTCGAA | *Delgado-Benito et al., 2018* |
| | gRif1- N2a (as above) | TGTGTGTACCAGGGCACTGT | This paper |
| | gRif1- N2b (as above) | ACTCTTAATGATACCATTCA | This paper |
| | gRev7-1 | CCTGATTCTCTATGTGCGCG | This paper |
| | gRev7-2 | GTGCGCGAGGTCTACCCGGT | This paper |
| CRISPR-Cas9 gene targeting in in-bulk cultures | gRev7-3 | CTATGTGCGCGAGGTCTACC | This paper |
| *Site-directed mutagenesis of knock-in template* | *PCR primers* | | |
| | **A1387 → S1387** | | |
| | Primer 1 | CAAATAGTAAATGAAGATAGTCAGGCTGCTGCGCTAGCCCC | This paper |
| | Primer 2 | GGGGCTAGGGCAGCAGCCTGACTATCTTCATTTACTATTTG | This paper |
| | **A1416 → S1416** | | |
| | Primer 1 | GATTCTTGCAGTGACAGCCAAGAGAGAGAGAGTGGTCAGC | This paper |
| | Primer 2 | GCTGACCACTCTCTCTCTCTTGGCTGTCACTGCAAGAATC | This paper |
| | **A1528 → S1528** | | |
| | Primer 1 | CGTTATCAAACAAGAAGAGCTTCGCAGGGTTTGATTTCTGC | This paper |
| | Primer 2 | GCAGAAATCAAACCCTGCGAAGCTCTTCTTGTTTGATAACG | This paper |

*Table 1 continued on next page*

*Table 1 continued*

| PCR primers | | |
| --- | --- | --- |
| *Brca1* – Nickase 1 and 2 clones | | |
| Fw | AAATGTGTGTGTGGAGCCATG | This paper |
| Rev | CTTCTCCAAACCAGTAGAGG | This paper |
| *Rif1* – Nickase 1 clones | | |
| Fw | GAGTAAATAAGCGCGAGCCG | *Delgado-Benito et al., 2018* |
| Rev | CGATCCGGAGTTAGTGGGTT | *Delgado-Benito et al., 2018* |
| *Rif1* – Nickase 2 clones | | |
| Fw | TTCCTTCCCTCAGTAGAG | This paper |
| Rev | GCAACAGGGCTGGCATTT | This paper |
| *Rif1$^{S\rightarrow A}$* – *Rif1* locus | | |
| Fw | GCGGTGCTTGAACTTCAGGG | This paper |
| Rev | GCTGCGTGCTCAGTCTCAAC | This paper |
| *Rif1$^{S\rightarrow A}$* – HR donor | | |
| Fw | TGTGGTGGCTCTGTTGCTGA | This paper |
| Rev | GCATGGTCACGAGCTTCACG | This paper |
| *Rif1$^{S1387A}$*, *Rif1$^{S1416A}$*, and *Rif1$^{S1528A}$* – *Rif1* locus | | |
| Analysis of genomic scars and knock-ins | Fw | ACTCTGAACCATACACTAGCAG | This paper |
| | Rev | TTGGGTGGAGCTTGCAGTGA | This paper |

Fw: forward; Rev: reverse.

37°C for 15 min to perform a hypotonic shock, and washed/fixed with 3:1 methanol (VWR)/glacial-acetic acid (Carl Roth) solution for 30 min at RT. Metaphase spreads were prepared by dropping fixed cells on humidified microscope slides, which were air-dried and placed at 42°C for 1 hr. Giemsa staining was performed by using KaryoMAX Giemsa Stain Solution and 1× Gurr Buffer (tablets, Gibco). Metaphases were acquired with the Automated Metaphase Finder System Metafer4 (MetaSystems).

## DNA fiber assay

Degradation of nascent DNA at stalled forks was assessed as follows. Exponentially growing CH12 cells were sequentially pulse-labeled with 40 µM of idoxuridine (IdU) (Sigma-Aldrich) and 400 µM of 5-chloro-2'-deoxyuridine thymidine (CldU) (Sigma-Aldrich) for exactly 20 min each, washed once with 1× PBS, and treated with 4 mM HU for 3 hr. Cells were collected and resuspended in 1× PBS at a concentration of $3.5 \times 10^5$ cells/ml. 3 µl of cell suspension was diluted with 10 µl of lysis buffer (200 mM Tris-HCl pH 7.5, 50 mM EDTA, and 0.5% [w/v] SDS) on a glass slide and incubated for 2 min at RT. The slides were titled at 15–60°, air-dried, and fixed with 3:1 methanol/acetic acid for 10 min. Slides were denatured with 2.5 M HCl for 80 min, washed with 1× PBS, and blocked with 5% BSA (Carl Roth) in PBS for 40 min. The newly replicated CldU and IdU tracks were labeled for 1.5 hr with anti-BrdU antibodies recognizing CldU (1:500, Abcam) and IdU (1:50, BD Biosciences), followed by 45 min incubation with secondary antibodies anti-mouse Alexa Fluor 488 (1:500, Invitrogen) and anti-rat Alexa Fluor 546 (1:500, Invitrogen). The incubations were performed in the dark in a humidified chamber. DNA fibers were visualized using a Carl Zeiss LSM800 confocal microscope at a 40× objective magnification, and images were analyzed using ImageJ software.

Whenever indicated, the DNA2 inhibitor NSC-105808 (*Kumar et al., 2017*) was added at a final concentration of 0.3 µM for 24 hr prior to HU addition.

## Proximity ligation assay (PLA)

Exponentially growing CH12 cells were incubated with 10 µM of 5-ethynyl-2'-deoxyuridine (EdU) (Merck) for 15 min. For each sample, cells were washed once with 1× PBS and split into two

aliquots, one of which was incubated with fresh media containing 4 mM HU for 3 hr. The other aliquot was incubated with fresh media without HU and processed for PLA in parallel (untreated condition). Cells were washed once with 1× PBS and fixed with 4% paraformaldehyde (Sigma-Aldrich) for 10 min at RT. Cells were washed twice with 1× PBS and then permeabilized for 5 min at RT using 0.2% Triton-X-100 (Roth). Cells were washed twice with 1× PBS and incubated for 30 min in the dark at RT with Click-iT Cell Reaction Buffer Kit (Thermo Fisher) supplemented with 25 µM biotin-azide (Thermo Fisher) according to the manufacturer's instructions to conjugate incorporated EdU with biotin. Specifically, 500 µl of the Click-iT reaction was used for 6 × $10^6$ cells. After the click reaction, cells were washed with 1× PBS and then blocked with 3% BSA in 1× PBS for 1 hr at 37°C in a humidified chamber. The blocking solution was removed and cells were incubated overnight at 4°C with primary antibodies against RIF1 (1:1000) and biotin (1:1000) in blocking solution. The following day the PLA was performed using the Duolink flowPLA Detection Kit – FarRed (Sigma-Aldrich) according to the manufacturer's instructions. Specifically, 40 µl of reaction mix per sample were used at each step, and all incubations were performed at 37°C in a humidified chamber. Cells were washed twice with Duolink wash buffer and incubated for 1 hr with the Duolink PLA probes anti-mouse plus (for biotin) and anti-rabbit minus (for RIF1) diluted 1:5 in blocking solution. Cells were then washed twice with Duolink wash buffer and incubated with Duolink ligation mix prepared by diluting ligation buffer 1:5 and ligase 1:40 in high-purity water for 30 min. Cells were washed twice and incubated for 100 min with the Duolink amplification mix prepared by diluting amplification buffer 1:5 and rolling circle polymerase 1:80 in high-purity water. Cells were washed twice and then incubated for 30 min with Duolink detection solution prepared by diluting detection buffer 1:5 in high-purity water. The detection solution was washed off, and cells were resuspended in 1× PBS containing 3% BSA. Samples were acquired on a BD LSRFortessa cell analyzer. To control for EdU incorporation, PLA values were expressed as the ratio of the mean fluorescent intensity of the HU-treated versus untreated conditions, which derived from the same EdU-incubated sample, as indicated above.

## Protein sequence analysis

The sequence alignment of RIF1 orthologs was performed simultaneously on the full-length proteins from all 18 species listed in *Figure 1—source data 1* using the multiple sequence alignment program Clustal Omega (clustalo version 1.2.4, https://www.ebi.ac.uk/Tools/msa/clustalo/).

The disorder profile plots were determined by the *Protein DisOrder prediction System* (PrDOS) server (https://prdos.hgc.jp/cgi-bin/top.cgi; *Ishida and Kinoshita, 2007*) using the template-based prediction option and with the prediction false-positive rate set to 5.0%.

## Statistical analysis

Information about the statistical analysis of the mass spectrometry datasets is included in the section 'Identification of RIF1 phosphoresidues' above. For all other data presented in this study, the statistical significance of differences between groups/datasets was determined by the Mann–Whitney *U*-test. Statistical details of experiments can be found in the figure legends.

## Acknowledgements

We thank all members of the Di Virgilio lab for their feedback and discussion; V Delgado-Benito (Di Virgilio lab, MDC, Berlin) for her contribution to the project development; L Keller (Di Virgilio lab, MDC, Berlin) for support with cloning, mutagenesis, and mice genotyping; C Brischetto (Scheidereit Lab, MDC, Berlin) for assistance with confocal microscopy; Aberdeen Proteomics facility (University of Aberdeen) for the mass spec analysis of Aph-induced hRIF1 phosphorylation; and the MDC FACS Core Facility and Dr. HP Rahn for support with cell sorting. Aliquots of ATRi and ATMi were generously provided by AG Henssen (MDC and ECRC, Berlin). Figures 1B and D, 2A, and 4C contain items created with BioRender.com. This work was supported by ERC grant 638897 (to MDV), the Helmholtz-Gemeinschaft Zukunftsthema 'Immunology and Inflammation' ZT-0027 (to MDV), P41 GM109824 and P41 GM103314 (to BTC), and Cancer Research UK awards C1445/A19059 and DRCPGM\100,013 (to ADD and SH).

## Additional information

### Competing interests

Matteo Andreani: is affiliated with Tacalyx GmbH. The author has no financial interests to declare. Javier Garzón: is affiliated with Adrestia Therapeutics Ltd. The author has no financial interests to declare. The other authors declare that no competing interests exist.

### Funding

| Funder | Grant reference number | Author |
|---|---|---|
| H2020 European Research Council | ERC Starting Grant 638897 | Michela Di Virgilio |
| Helmholtz Association | Helmholtz-Gemeinschaft Zukunftsthema "Immunology and Inflammation" ZT-0027 | Michela Di Virgilio |
| National Institutes of Health | P41 GM109824 | Brian T Chait |
| National Institutes of Health | P41 GM103314 | Brian T Chait |
| Cancer Research UK | C1445/A19059 | Shin-ichiro Hiraga Anne D Donaldson |
| Cancer Research UK | DRCPGM\100013 | Shin-ichiro Hiraga Anne D Donaldson |
| MDC | | Michela Di Virgilio |

The funders had no role in study design, data collection and interpretation, or the decision to submit the work for publication.

### Author contributions

Sandhya Balasubramanian, Conceptualization, Formal analysis, Investigation, Methodology, Resources, S.B. and M.A. contributed the majority of experiments, Writing – review and editing; Matteo Andreani, Conceptualization, Formal analysis, Investigation, Methodology, Resources; Júlia Goncalves Andrade, Tannishtha Saha, Devakumar Sundaravinayagam, Javier Garzón, Daniel B Rosen, Investigation; Wenzhu Zhang, Formal analysis, Investigation, W.Z. analysed the I-DIRT proteomic datasets and identified RIF1 phosphoresidues in the IDR-CII SQ cluster; Oliver Popp, Formal analysis, Investigation; Shin-ichiro Hiraga, Formal analysis, Investigation, S.H. performed the mass spec experiment for the aphidicolin-induced hRIF1 phosphorylation; Ali Rahjouei, Investigation, Visualization; Philipp Mertins, Supervision; Brian T Chait, Anne D Donaldson, Supervision, Writing – review and editing; Michela Di Virgilio, Conceptualization, Formal analysis, Funding acquisition, Investigation, Project administration, Supervision, Visualization, Writing – original draft, Writing – review and editing

### Author ORCIDs

Sandhya Balasubramanian http://orcid.org/0000-0003-1830-5737
Matteo Andreani http://orcid.org/0000-0003-1426-5854
Shin-ichiro Hiraga http://orcid.org/0000-0002-8722-3869
Daniel B Rosen http://orcid.org/0000-0002-0412-6239
Anne D Donaldson http://orcid.org/0000-0001-7842-8136
Michela Di Virgilio http://orcid.org/0000-0001-5189-0793

### Ethics

All experiments were performed in compliance with the European Union (EU) directive 2010/63/EU, and in agreement with Landesamt für Gesundheit und Soziales directives (LAGeSo, Berlin, Germany). Mice were kept in a specific pathogen-free (SPF) barrier facility under standardized conditions (20+/-2 °C temperature; 55%±15% humidity) on a 12 h light/12 h dark cycle. Animals were maintained in small groups (4 to 5) or as breeding pairs in individually ventilated cages to ensure optimal habitat

condition. Mice were maintained on a C57BL/6 background and animals of both genders were used for the experiments.

## Decision letter and Author response
Decision letter https://doi.org/10.7554/eLife.75047.sa1
Author response https://doi.org/10.7554/eLife.75047.sa2

## Additional files

### Supplementary files
• Transparent reporting form

### Data availability
The mass spectrometry proteomics data have been deposited to Zenodo (RIF1 I-DIRT, DOI: https://doi.org/10.5281/zenodo.5643859), and to the ProteomeXchange Consortium via the PRIDE partner repository with the dataset identifiers PXD031972 (HU-induced phosphorylation of RIF1 in mouse B lymphocytes) and PXD032015 (Aph-induced hRIF1 phosphorylation). Source Data files have been provided for all images of gels/blots/metaphases/fibers in main and supplementary figures and for MaxQuant analysis output of the RIF1 I-DIRT interactome list relative to this study (Figure 1). All other data generated during this study are included in the manuscript and supporting files.

The following datasets were generated:

| Author(s) | Year | Dataset title | Dataset URL | Database and Identifier |
|---|---|---|---|---|
| Zhang W, Chait BT | 2021 | LC-MS raw data for RIF1 complexes and GPM database search results | https://doi.org/10.5281/zenodo.5643859 | Zenodo, 10.5281/zenodo.5643859 |
| Popp O, Mertins P | 2022 | HU-induced phosphorylation of RIF1 in mouse B lymphocytes | https://www.ebi.ac.uk/pride/PXD031972 | PRIDE, PXD031972 |
| Hiraga S | 2022 | Aph-induced hRIF1 phosphorylation | https://www.ebi.ac.uk/pride/PXD032015 | PRIDE, PXD032015 |

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

# Appendix 1

## Appendix 1—key resources table

| Reagent type (species) or resource | Designation | Source or reference | Identifiers | Additional information |
|---|---|---|---|---|
| Genetic reagent (*Mus musculus*) | *Rif1*$^{FH/FH}$ and *Rif1*$^{F/F}$*Cd19*$^{Cre/+}$ mice | *Cornacchia et al., 2012*; *Di Virgilio et al., 2013* | | |
| Biological sample (*M. musculus*) | Primary mouse embryonic fibroblasts | This paper | | Isolated from *WT* and *Rif1*$^{FH/FH}$ mice |
| Biological sample (*M. musculus*) | Primary splenocytes | This paper | | Isolated from *WT*, *Rif1*$^{FH/FH}$ and *Rif1*$^{F/F}$*Cd19*$^{Cre/+}$ mice |
| Gene (*M. musculus*) | RIF1 | UniProt | Q6PR54-1 | |
| Strain, strain background (*Escherichia coli*) | Stbl3 (HB101) | Thermo Fisher | C737303 | Chemically competent cells |
| Cell line (*M. musculus*) | CH12 | *Nakamura et al., 1996* | | |
| Cell line (*M. musculus*) | *Brca1*$^{mut}$ CH12 | This paper | | See *Figure 2—figure supplement 1* and Materials and methods |
| Cell line (*M. musculus*) | *Brca1*$^{mut}$*Rif1*$^{-/-}$ CH12 | This paper | | See *Figure 2—figure supplement 2* and Materials and methods |
| Cell line (*M. musculus*) | *Brca1*$^{mut}$ *Rif1*$^{S→A}$ CH12 | This paper | | See *Figure 2—figure supplement 2* and Materials and methods |
| Cell line (*M. musculus*) | *Rif1*$^{S→A}$ CH12 | This paper | | See *Figure 2—figure supplement 2*, *Figure 3—figure supplement 1*, and Materials and methods |
| Cell line (*M. musculus*) | *Rif1*$^{S1387A}$ CH12 | This paper | | See *Figure 3—figure supplement 1* and Materials and methods |
| Cell line (*M. musculus*) | *Rif1*$^{S1416A}$ CH12 | This paper | | See *Figure 3—figure supplement 1* and Materials and methods |
| Cell line (*M. musculus*) | *Rif1*$^{S1528A}$ CH12 | This paper | | See *Figure 3—figure supplement 1* and Materials and methods |
| Transfected construct (*M. musculus*) | pMA-Rif1$^{S→A}$ (pMA is a GeneArt Cloning Vector from Life Technologies) | This paper | | HR donor plasmid for introducing Ser→Ala mutations at S1387, S1416, and S1528 of mouse RIF1 |
| Transfected construct (*M. musculus*) | pMA-Rif1$^{S1387A}$ | This paper | | HR donor for introducing Ser→Ala mutations at S1387 of mouse RIF1 |
| Transfected construct (*M. musculus*) | pMA-Rif1$^{S1416A}$ | This paper | | HR donor for introducing Ser→Ala mutations at S1416 of mouse RIF1 |
| Transfected construct (*M. musculus*) | pMA-Rif1$^{S1528A}$ | This paper | | HR donor for introducing Ser→Ala mutations at S1528 of mouse RIF1 |
| Chemical compound, drug | αCD40 | BioLegend | 102902 | |
| Chemical compound, drug | IL-4 | Sigma-Aldrich | I1020-5UG | |
| Chemical compound, drug | TGFβ | R&D Systems | 7666MB-005/CF | |
| Chemical compound, drug | Olaparib (PARPi) | Selleckchem.com | S1060 SEL-S1060-10MM | |

*Appendix 1 Continued on next page*

*Appendix 1 Continued*

| Reagent type (species) or resource | Designation | Source or reference | Identifiers | Additional information |
|---|---|---|---|---|
| Chemical compound, drug | Idoxuridine (IdU) | Sigma-Aldrich | I0050000 | |
| Chemical compound, drug | 5-Chloro-2'-deoxyuridine thymidine (CldU) | Abcam | ab213715 | |
| Chemical compound, drug | Hydroxyurea (HU) | Sigma-Aldrich | H8627-5G | |
| Chemical compound, drug | 5-Ethynyl-2'-deoxyuridine (EdU) | Merck | 900584–50MG | |
| Chemical compound, drug | Dynabeads Protein A | Thermo Fisher | 10001D | |
| Antibody | FLAG-M2 (mouse monoclonal) | Sigma-Aldrich | F3165 | WB (1:1000) |
| Antibody | FLAG-M2 peroxidase (HRP) (mouse monoclonal) | Sigma-Aldrich | A8592-.2MG | WB (1:1000) |
| Antibody | RIF1 (rabbit polyclonal) | *Di Virgilio et al., 2013* | | WB (1:2500) PLA (1:1000) |
| Antibody | PP1 (rabbit polyclonal) | Abcam | ab137512 | WB (1:1000) |
| Antibody | Vinculin (mouse monoclonal) | Sigma-Aldrich | V9131 | WB (1:10,000) |
| Antibody | β-Actin (mouse monoclonal) | Sigma-Aldrich | A5441 | WB (1:10,000) |
| Antibody | Tubulin (rabbit polyclonal) | Abcam | ab4074 | WB (1:10,000) |
| Antibody | RPA32 (mouse monoclonal) | Millipore | NA19L | WB (1:1000) |
| Antibody | pRPA32 (S4/S8) (rabbit polyclonal) | Bethyl Laboratories | A300-245A-M | WB (1:1000) |
| Antibody | pCHK1 (S345) (rabbit monoclonal) | Cell Signaling | 2348S | WB (1:1000) |
| Antibody | CHK1 (mouse monoclonal) | Cell Signaling | 2360S | WB (1:1000) |
| Antibody | Phospho-Histone H2A.X (S139) (mouse monoclonal) | Millipore | 05-636 | WB (1:1000) |
| Antibody | H2AX (rabbit polyclonal) | Novus Biologicals | NB100-383 | WB (1:1000) |
| Antibody | HRP-goat anti-rabbit heavy chain (goat polyclonal) | Jackson ImmunoResearch | 111-035-008 | WB (1:10,000) |
| Antibody | HRP-goat anti-mouse heavy chain (goat polyclonal) | Jackson ImmunoResearch | 115-035-008 | WB (1:10,000) |
| Antibody | HRP-mouse anti-rabbit light chain (mouse monoclonal) | Jackson ImmunoResearch | 211-032-171 | WB (1:10,000) |
| Peptide, recombinant protein | Biotin-azide | Thermo Fisher | B10184 | 25 µM |
| Commercial assay or kit | Click-iT Cell Reaction Buffer Kit | Thermo Fisher | C10269 | 500 µl |

*Appendix 1 Continued on next page*

*Appendix 1 Continued*

| Reagent type (species) or resource | Designation | Source or reference | Identifiers | Additional information |
|---|---|---|---|---|
| Commercial assay or kit | Duolink flowPLA Mouse/Rabbit Starter Kit - Far Red | Sigma-Aldrich/Duolink | DUO94104 | |
| Software | MacVector | https://macvector.com/ | RRID:SCR_015700 | |
| Software | FlowJo | https://www.flowjo.com/ | RRID:SCR_008520 | |
| Software | ImageJ | https://imagej.nih.gov/ij/ | RRID:SCR_003070 | |
| Software | GPM software | *Beavis, 2006* | | |

