## [Editor Report]

This paper reports a novel regulatory mechanism that modulates RIF1 function during the DNA replication stress response. The authors identify three residues within the mouse RIF protein that can be phosphorylated in an ATM/ATR-dependent manner. Interestingly, this phosphorylation is dispensable for the ability of RIF1 to limit double-strand break resection but is required to protect stalled replication forks. The results are of clear interest for the field of DNA replication and repair.

---

## [Decision Letter]

**Decision letter after peer review:**

Thank you for submitting your article "Protection of nascent DNA at stalled replication forks is mediated by phosphorylation of RIF1 intrinsically disordered region" for consideration by *eLife*. Your article has been reviewed by 2 peer reviewers, and the evaluation has been overseen by a Reviewing Editor and Jessica Tyler as the Senior Editor. The reviewers have opted to remain anonymous.

Essential revisions:

1) The statement that the phosphorylation of the three Ser residues in RIF1 is a molecular switch between its cellular functions should be toned down in the manuscript, as the genetic backgrounds for the activities analysed are not equivalent.

2) The phenotype of single mutants should be shown; are all three mutations necessary to achieve the reported effect on fork protection?

3) Specificity of DNA2 chemical inhibition should be strengthened with a second inhibitor or with genetic data.

4) Ideally ATM/ATR dependence of the phosphorylation events should be demonstrated.

5) The authors should include new experiments to investigate how blocking phosphorylation of the SQ cluster prevents RIF1 function at replication forks. Does RIF1 phosphorylation regulate the interaction of RIF1 with the PP1 phosphatase, on the basis of previous studies suggesting that this interaction is essential for the role of RIF1 in replication fork protection (Garzon et al., Cell Reports 2019)? Along the same line, does suppression of RIF1 phosphorylation prevent the ability of RIF1 to limit WRN phosphorylation and DNA2 activation?

6) Does RIF1 phosphorylation affect the interaction of RIF1 with stalled replication forks?

7) The experiments of Figure 2 show that loss of RIF1 reduces the accumulation of chromatid breaks and radial chromosomes in BRCA1 mutant cells treated with the PARP inhibitor Olaparib. The same reduction was not observed in cells expressing the RIF1 S to A mutants. On the basis of these findings, the authors conclude that phosphorylation of the SQ cluster is dispensable for the ability of RIF1 to inhibit DSB resection at collapsed replication forks. However, the experiments included in Figure 2 do not provide any direct readout for DSB resection or HR efficiency. Additional experiments that directly monitor DSB resection (e.g., by monitoring RPA or ssDNA accumulation) need to be included to support the authors' conclusion.

8) The three RIF1 phosphorylation sites were identified through a mass spectrometry experiment in irradiated mouse B cells. The authors should test whether the same residues are phosphorylated upon PARP inhibition or HU treatment to support the follow-up experiments included in Figures 2 and 3.

*Reviewer #1 (Recommendations for the Authors):*

Figure 2 (and S2) describes generation of CH12 cell lines carrying Brca1 mutation and/or Rif1 deletion. Both single- and double-mutant cells recapitulate the known responses to PARPi treatment (viability, radial chromosomes) and, as predicted, undergo CSR only when Rif1 is wt. CH12 cells are a cellular model superior to B cells isolated from transgenic mice, because they can be easily engineered in vitro. This technical information could be useful for the field.

Figure 2G-I and Figure 3: S to A mutations are introduced concomitantly at the three RIF1 SQ sites mentioned above, in the Brca1 mutant and wt cell lines. RIF1 triple mutant behaves as wt RIF1 in radial chromosomes and viability assays after PARPi treatment, as well as in ability to activate CSR, all performed in Brca1mut cells. The triple mutant phenotype however switches to Rif1-/-, when replication fork protection is assessed in Brca1wt cells. RIF1 is known to be required for fork protection (in BRCA1 wt cells) and mutation of the three residues abrogates protection.

The cellular context is, however, not equivalent: the PARPi-induced responses are assessed in Brca1mut background and fork protection in Brca1wt cells. As such, it cannot be said that phosphorylation of the three residues represents a molecular switch between the two functions.

Fork degradation in Rif1-/- cells is DNA2 dependent, and the paper shows the same dependence in the mutant carrying the three RIF1 S to A alterations. A chemical DNA2 inhibitor is used in Figure 3C,D. Given the potential non-specific activities of such inhibitors it would be useful to perform specificity assays (e.g. genetic DNA2 inhibition) to strengthen these data. In addition, evidence that RIF1 phosphorylation at these sites is indeed ATM- or ATR-dependent should ideally be provided.

---

## [Author Response]

Essential revisions:1) The statement that the phosphorylation of the three Ser residues in RIF1 is a molecular switch between its cellular functions should be toned down in the manuscript, as the genetic backgrounds for the activities analysed are not equivalent.

As suggested by the Reviewers, we have modified the manuscript to clarify that our study reports a key molecular determinant of RIF1 role specifically in DNA replication fork protection rather than a molecular switch between its various cellular functions.

2) The phenotype of single mutants should be shown; are all three mutations necessary to achieve the reported effect on fork protection?

To answer the Reviewer’s question, we have first generated CH12 cell lines bearing single amino acid substitutions at each of the three SQ phosphoresidues in the cluster (S1387A, S1416A, and S1528A). Next, we assessed the capability of RIF1 single SQ mutants to protect nascent DNA at stalled forks *via* the DNA fiber assay. Interestingly, we found that while *Rif1^S1387A^* and *Rif1^S1416A^* cell lines were proficient in protecting stalled forks from degradation, *Rif1^S1528A^* cells exhibited a reproduceable fork degradation defect. However, the phenotype was modest and did not recapitulate the severe defect of RIF1 knock-out and cluster S→A mutant cells. Altogether, this data suggests that phosphorylation of S1528 contributes to, but is not sufficient for, fork protection, and that multiple phosphorylation events within the IDR-CII SQ cluster are responsible for RIF1 ability to protect nascent DNA under conditions of replication stress. These new results have been summarized in Figure 3, panels E and F (nascent fork degradation analysis), and Figure 3 —figure supplement 1, panels F and G (characterization of the newly-generated single SQ mutant cell lines) of the revised manuscript.

3) Specificity of DNA2 chemical inhibition should be strengthened with a second inhibitor or with genetic data.

We agree with the Reviewers that chemical inhibition of enzymatic activities does not guarantee the specificity that deletion or downregulation of the enzymes themselves provides, nor do these approaches necessarily yield the same outcomes. However, the dependency of the fork degradation phenotype of RIF1-deficient cells on DNA2 has been assessed in two earlier studies, which both employed the same inhibitor of DNA2 nuclease activity that we used (DNA2i NSC-105808, Kumar et al., *Oncogenesis* 2017) in addition to RNA interference-based experiments (Mukherjee et al., *Nat Commun* 2019; Garzón et al., *Cell Rep* 2019). In these studies, downregulation of DNA2 and chemical inhibition of its nuclease activity returned the same results and unambiguously proved that DNA2 is the primary nuclease responsible for degradation of stalled forks in the absence of RIF1 (Mukherjee et al., *Nat Commun* 2019; Garzón et al., *Cell Rep* 2019). Given this now well-established mechanistic relationship, in our study we decided to assess for protection of stalled forks under condition of DNA2 inhibition to provide an additional confirmation of the fork degradation phenotype of RIF1 SQ cluster mutants. For these reasons, we believe that repeating the experiments with a second inhibitor or genetic approaches would not provide any new insight into the underlying mechanism, and respectfully decided not to pursue this set of experiments.

4) Ideally ATM/ATR dependence of the phosphorylation events should be demonstrated.

To address this point, we have optimized RIF1 pull-downs for the identification of post-translational modifications in primary B cells, and compared the RIF1 peptide composition of mock- *versus* HU-treated samples in the absence and presence of ATM (AZD0156) or ATR (BAY 1895344) inhibitors. Interestingly, the only SQ site that was identified to be phosphorylated in a HU-dependent manner was indeed one of the three conserved motifs of the IDR-CII SQ cluster (S1416). In addition, HU-induced phosphorylation of S1416 was reduced following treatment with the ATM inhibitor and showed a similar, albeit not significant, trend with the ATRi-treatment as well. These results are included in the new Figure 4 (panels C and D) and build on the initial identification of S1416 in the RIF1 I-DIRT preparations by demonstrating the ATM/ATR-dependency of HU-induced phosphorylation. Although this new dataset did not yield additional phosphorylated SQ motifs, we cannot exclude the likely possibility that peptides containing phospho-S1387 and phospho-S1528 residues might simply be undetectable under the conditions employed for this new set of pull-downs and mass spectrometry analysis, as this is a frequent occurrence with proteomic-based discovery approaches.

5) The authors should include new experiments to investigate how blocking phosphorylation of the SQ cluster prevents RIF1 function at replication forks. Does RIF1 phosphorylation regulate the interaction of RIF1 with the PP1 phosphatase, on the basis of previous studies suggesting that this interaction is essential for the role of RIF1 in replication fork protection (Garzon et al., Cell Reports 2019)? Along the same line, does suppression of RIF1 phosphorylation prevent the ability of RIF1 to limit WRN phosphorylation and DNA2 activation?

To mechanistically dissect how phosphorylation of the conserved intrinsically disordered region (IDR) cluster contributes to RIF1 role in protection of stalled replication forks, we proceeded with two lines of investigation.

First, as suggested by the Reviewers, we monitored RIF1-PP1 interaction in control and RIF1^S→A^-expressing cells *via* co-immunoprecipitation studies. We found that RIF1 mutant protein retains the ability to interact with PP1, thus indicating that the abrogation of phosphorylation events in the conserved cluster does not have a major impact on RIF1-PP1 association. This result is presented in a new Figure 4, panel A, in the revised manuscript.

Next, we assessed the capability of *Rif1^S→A^* CH12 cells to recruit mutant RIF1 to stalled replication forks. To do so, we applied a proximity ligation assay (PLA) that employs flow cytometry measurements to quantitatively assess the localization of RIF1 at sites of EdU incorporation in the presence and absence of HU treatment. This approach adapted the commonly used PLA-based approach for quantitative analysis of protein interactions with nascent DNA (Roy et al., *J Cell Biol* 2018) by replacing the final microscopy-based detection step with flow cytometry analysis. As a consequence, it quantitates RIF1 binding to replicated DNA as mean RIF1-EdU proximity signal intensity over the entire cell population. We found that while RIF1 and EdU co-localization increased as expected upon HU exposure in control cell lines, RIF1-EdU proximity signal was only modestly affected by HU treatment in *Rif1^S→A^* clonal derivatives. This data indicates that recruitment of RIF1^S*→*A^ mutant to stalled replication forks is not as efficient as for the wild-type protein counterpart. The results are summarized in the new Figure 4, panel B.

Although we cannot rule out the possibility that some aspects of RIF1 activity at the forks might also be influenced, these new findings suggest that phosphorylation of the IDR-CII SQ cluster facilitates RIF1 recruitment to, rather than its function at, stalled replication forks.

6) Does RIF1 phosphorylation affect the interaction of RIF1 with stalled replication forks?

As indicated in our response to point 5 above, we compared the binding of wild-type and S→A cluster mutant RIF1 proteins to newly-replicated DNA in the absence and presence of HU, and found that phosphorylation of RIF1 IDR-CII SQ cluster promotes RIF1 interaction with stalled replication forks (new Figure 4B). The key players and precise molecular mechanism underlying phosphorylation-dependent recruitment of RIF1 to replicated DNA will be the object of future studies.

7) The experiments of Figure 2 show that loss of RIF1 reduces the accumulation of chromatid breaks and radial chromosomes in BRCA1 mutant cells treated with the PARP inhibitor Olaparib. The same reduction was not observed in cells expressing the RIF1 S to A mutants. On the basis of these findings, the authors conclude that phosphorylation of the SQ cluster is dispensable for the ability of RIF1 to inhibit DSB resection at collapsed replication forks. However, the experiments included in Figure 2 do not provide any direct readout for DSB resection or HR efficiency. Additional experiments that directly monitor DSB resection (e.g., by monitoring RPA or ssDNA accumulation) need to be included to support the authors' conclusion.

We agree with the Reviewers that although the accumulation of aberrations in BRCA1-mutated cells (chromatid breaks and radial chromosomes) and the efficiency of CSR in B cells are both impacted by defects in the regulation of DNA end processing, they do not represent a direct read-out for it. To directly assess whether phosphorylation of the IDR-CII SQ cluster modulates RIF1 role in the inhibition of end resection, we compared RPA (RPA32 S4/S8) phosphorylation levels following ionizing irradiation (IR)-induced DNA damage in RIF1-proficient, -deficient, and RIF1^S→A^-expressing cells. Both in wild-type and BRCA1-mutant backgrounds, IR induced a marked phosphorylation of RPA in *Rif1^-/-^*, but not *Rif1^S→A^*, cells thus confirming our initial conclusion that RIF1 function in DSB end protection is not influenced by the phosphorylation status of the IDR-CII SQ cluster. This new data is presented in Figure 2 —figure supplement 2H (BRCA1-mutated background) and figure 3 —figure supplement 1E (WT background).

8) The three RIF1 phosphorylation sites were identified through a mass spectrometry experiment in irradiated mouse B cells. The authors should test whether the same residues are phosphorylated upon PARP inhibition or HU treatment to support the follow-up experiments included in Figures 2 and 3.

As indicated in our response to point 4 above, we have performed mass spectrometry-based analysis of RIF1 purified from primary B cells after treatment with HU. S1416 was indeed found to be phosphorylated in HU(as well as ATM and potentially ATR)-dependent manner (new Figure 4, C and D), which is in agreement with the initial interpretation of our study. Although this new dataset did not yield additional phosphorylated SQ motifs, we cannot exclude the likely possibility that peptides containing phospho-S1387 and phospho-S1528 residues might be simply undetectable under the specific conditions employed for this new set of primary B cell pull-downs and mass spectrometry analysis. In support of this point, an independent proteomics analysis of hRIF1 isolated from Flp-In T-REx GFP-RIF1-L cells identified S1542 (which corresponds to S1528 in mouse RIF1, see Figure 1 – source data 2) as a SQ site phosphorylated following treatment with the DNA polymerase inhibitor aphidicolin (new Figure 4E). Collectively, these results build on the initial identification of S1416 and S1528 in the RIF1 I-DIRT preparations (Figure 1, E and F), and demonstrate that SQ motifs in the conserved cluster are phosphorylated following treatment with replication stress-inducing agents.

Reviewer #1 (Recommendations for the Authors):Figure 2 (and S2) describes generation of CH12 cell lines carrying Brca1 mutation and/or Rif1 deletion. Both single- and double-mutant cells recapitulate the known responses to PARPi treatment (viability, radial chromosomes) and, as predicted, undergo CSR only when Rif1 is wt. CH12 cells are a cellular model superior to B cells isolated from transgenic mice, because they can be easily engineered in vitro. This technical information could be useful for the field.

We thank the Reviewer for highlighting that the approach used in our study could be useful for the field. We have indeed already received expression of interests from other laboratories for the cell lines we generated, and we will gladly share all reagents and any technical tip alongside the published protocols.

Figure 2G-I and Figure 3: S to A mutations are introduced concomitantly at the three RIF1 SQ sites mentioned above, in the Brca1 mutant and wt cell lines. RIF1 triple mutant behaves as wt RIF1 in radial chromosomes and viability assays after PARPi treatment, as well as in ability to activate CSR, all performed in Brca1mut cells. The triple mutant phenotype however switches to Rif1-/-, when replication fork protection is assessed in Brca1wt cells. RIF1 is known to be required for fork protection (in BRCA1 wt cells) and mutation of the three residues abrogates protection.The cellular context is, however, not equivalent: the PARPi-induced responses are assessed in Brca1mut background and fork protection in Brca1wt cells. As such, it cannot be said that phosphorylation of the three residues represents a molecular switch between the two functions.

As indicated in our response to point 1 of the *Essential Revisions* section, we have modified the manuscript to clarify that our study reports a key molecular determinant of RIF1 role specifically in DNA replication fork protection rather than a molecular switch between its various cellular functions.

Fork degradation in Rif1-/- cells is DNA2 dependent, and the paper shows the same dependence in the mutant carrying the three RIF1 S to A alterations. A chemical DNA2 inhibitor is used in Figure 3C,D. Given the potential non-specific activities of such inhibitors it would be useful to perform specificity assays (e.g. genetic DNA2 inhibition) to strengthen these data. In addition, evidence that RIF1 phosphorylation at these sites is indeed ATM- or ATR-dependent should ideally be provided.

As indicated in our response to point 3 of the *Essential Revisions* section, the dependency of the fork degradation phenotype of RIF1-deficient cells on DNA2 has been extensively investigated by two earlier studies, which both employed downregulation of DNA2 in parallel to chemical inhibition of its nuclease activity (Mukherjee et al., *Nat Commun* 2019; Garzón et al., *Cell Rep* 2019). In our study, we have exploited this now well-established mechanistic relationship to provide an additional confirmation of the fork degradation phenotype of RIF1 SQ cluster mutants. For these reasons, we believe that repeating the experiments with a second inhibitor or genetic approaches would not provide any new insight into the underlying mechanism, and respectfully decided not to pursue this set of experiments.

As indicated in our responses to points 4 and 8 of the *Essential Revisions* section, we started to address the ATM/ATR dependency of the phosphorylation events in the cluster by performing mass spectrometry analysis of RIF1 isolated from primary B cells. The IDR-CII cluster motif S1416Q was indeed identified as a HU-induced phosphosite, and the phosphorylation was reduced following treatment with ATM, and to a lesser extent ATR, inhibitors.